# IMPLICIT BIAS OF GRADIENT DESCENT BASED ADVERSARIAL TRAINING ON SEPARABLE DATA

**Yan Li, Huan Xu, Tuo Zhao**
H. Milton Stewart School of Industrial and Systems Engineering
Georgia Institute of Technology
Atlanta, GA 30318
`{yli939, huan.xu, tourzhao}@gatech.edu`

**Ethan X.Fang**
Department of Statistics
Pennsylvania State University
University Park, PA 16802
`xxf13@psu.edu`

## ABSTRACT

Adversarial training is a principled approach for training robust neural networks. Despite of tremendous successes in practice, its theoretical properties still remain largely unexplored. In this paper, we provide new theoretical insights of gradient descent based adversarial training by studying its computational properties, specifically on its implicit bias. We take the binary classification task on linearly separable data as an illustrative example, where the loss asymptotically attains its infimum as the parameter diverges to infinity along certain directions. Specifically, we show that for any fixed iteration $T$, when the adversarial perturbation during training has proper bounded $\ell_2$-norm, the classifier learned by gradient descent based adversarial training converges in direction to the maximum $\ell_2$-norm margin classifier at the rate of $\widetilde{\mathcal{O}}(1/\sqrt{T})$, significantly faster than the rate $\mathcal{O}\left(1/\log T\right)$ of training with clean data. In addition, when the adversarial perturbation during training has bounded $\ell_q$-norm with $q \geq 1$, the resulting classifier converges in direction to a maximum mixed-norm margin classifier, which has a natural interpretation of robustness, as being the maximum $\ell_2$-norm margin classifier under worst-case $\ell_q$-norm perturbation to the data. Our findings provide theoretical backups for adversarial training that it indeed promotes robustness against adversarial perturbation.

## 1 INTRODUCTION

Deep neural networks have achieved remarkable success on various tasks, including visual and speech recognitions, with intriguing generalization abilities to unseen data (Krizhevsky et al., 2012; Hinton et al., 2012). One salient feature of deep models is its overparameterization, with the number of parameters several orders of magnitude larger than the training sample size. As a consequence of such overparameterization, it is likely that the empirical loss function, in addition to being non-convex, can have substantial amount of global minimizers (Choromanska et al., 2015), while only a small subset of global minimizers have the desired generalization properties (Brutzkus et al., 2018).

Contrary to the worst-case reasoning above, researchers have observed that simple first-order algorithm such as Stochastic Gradient Descent (SGD) [1], performs surprisingly well in practice, even without any explicit regularization terms in the objective function (Zhang et al., 2017). Inspired by classical computational learning theories, one plausible explanation of such a remarkable phenomenon is that the training algorithm enjoys some implicit bias. That is, the training algorithm tends to converge to certain kinds of solutions (Neyshabur et al., 2015b;c), and SGD converges to low-capacity solutions with the desired generalization property (Brutzkus et al., 2018). Recently, some exciting works have related the implicit bias to specific first-order algorithms (Wilson et al.,

---

[1]In conjunction with Dropout (Srivastava et al., 2014) and Batch Normalization (Ioffe and Szegedy, 2015)

2017), stopping time (Hoffer et al., 2017), and optimization geometry (Gunasekar et al., 2018a; Keskar et al., 2017). Some practical suggestions based on these findings have also been proposed to further improve the generalization ability of deep networks (Neyshabur et al., 2015a).

Despite the aforementioned phenomenal success achieved by deep neural networks, it is observed that adversarially constructed small perturbation to the input can potentially fool the network into making wrong predictions with high confidence (Szegedy et al., 2014; Goodfellow et al., 2015). This issue raises serious concerns about using neural network for some security-sensitive tasks (Papernot et al., 2017). Researchers have devised various mechanisms to generate and defend against adversarial perturbations (Goodfellow et al., 2015; Moosavi-Dezfooli et al., 2016; Carlini and Wagner, 2017; Athalye et al., 2018; Xie et al., 2018; Papernot et al., 2016). However, most of the defense mechanisms are heuristic or ad-hoc, which lack principled theoretical justification (Carlini and Wagner, 2016; He et al., 2017). Inspired by literatures in robust optimization (Wald, 1939; Ben-Tal et al., 2009), Feige et al. (2015); Madry et al. (2018) formalize the notion of achieving adversarial robustness (i.e., having small adversarial risk) as solving the following minimax optimization problem

$$\min_{\theta \in \mathbb{R}^d} \mathcal{L}_{\mathrm{adv}}^{\mathrm{E}}(\theta) = \min_{\theta \in \mathbb{R}^d} \mathbb{E}_{(x,y) \sim \mathcal{D}} \Big[ \max_{\delta \in \Delta} \ell(\theta, x + \delta, y) \Big], \tag{1}$$

where $\Delta$ is the set that each sample could be contaminated by arbitrary perturbation chosen within this set. As a common practice, adversarial training refers to the finite-sample empirical version of (1) without access to the underlying distribution $\mathcal{D}$ that

$$\min_{\theta \in \mathbb{R}^d} \mathcal{L}_{\mathrm{adv}}(\theta) = \min_{\theta \in \mathbb{R}^d} \sum_{i=1}^{N} \max_{\delta_i \in \Delta} \ell(\theta, x_i + \delta, y_i). \tag{2}$$

A commonly adopted approach to solving (2) is the the Gradient Descent based Adversarial Training (GDAT) method. At each iteration, GDAT first solves the inner maximization problem (approximately) for adversarial perturbations, and then uses the gradient of the loss function evaluated at the perturbed samples to perform a gradient descent step on the parameter $\theta$. A natural question is then how adversarial training helps the trained model in achieving adversarial robustness. Some recent theoretical results partially answer this question, such as deriving adversarial risk bound (Athalye et al., 2018), relating it to the distributionally robust optimization (Sinha et al., 2018), and characterizing trade-offs between robustness and accuracy via regularization (Zhang et al., 2019).

Yet, all existing results neglect the algorithmic effect during the training process in promoting adversarial robustness. Inspired by the significant role of algorithmic bias in the generalization of neural networks, it is natural to ask

> ***Does gradient descent based adversarial training enjoy any implicit bias property?***
> ***If so, does the implicit bias provide insights on how adversarial training promotes robustness?***

Motivated by these questions, in this paper, we study the algorithmic effect of adversarial training by investigating the implicit bias of GDAT. Due to current technical limits in directly analyzing deep neural networks, we analyze a simpler model, with the key characteristics that the model overfits the training data while being able to generalize well. Specifically, we take the binary classification with linearly separable data as an example. This helps us focus on the effect of implicit bias without dealing with complicated structures of neural networks.

**Main Contributions.** We summarize our main theoretical findings below.

● Our first part of result shows an interesting interplay between adversarial perturbation and implicit bias of the gradient descent (GD). By exploiting this interplay, we show a property of adversarial training that is not known in the literature before: adversarial training accelerates convergence. Specifically, when the perturbation is bounded by $\ell_2$-norm, i.e., $\Delta = \{\delta \in \mathbb{R}^d : ||\delta||_2 \leq c\}$, with proper choice of $c$, the gradient descent based adversarial training is directionally convergent that $\lim_{t \to \infty} \frac{\theta^t}{||\theta^t||_2} = u_2$, where $u_2$ is the maximum $\ell_2$-norm margin hyperplane (i.e., standard SVM) of the training data. In addition, when the perturbation level $c$ is set according to $T$ appropriately, the rate of convergence is $\widetilde{\mathcal{O}}(1/\sqrt{T})^2$, which is exponentially faster than the rate $\mathcal{O}(1/\log T)$ when we use standard clean training, i.e., training with clean data using gradient descent. Based on this, we establish that the convergence of training loss on clean data using GDAT is almost exponentially faster than standard clean training using GD.

---

[2]$\widetilde{\mathcal{O}}$ hides logarithmic factor.

• Our second part of result shows that adversarial training adapts the implicit bias of gradient descent for different adversarial perturbation geometry. Specifically, when the perturbation is bounded by $\ell_q$-norm for $q \geq 1$, i.e., $\Delta = \{\delta \in \mathbb{R}^d : ||\delta||_q \leq c\}$, with proper choice of $c$, the gradient descent based adversarial training is directionally convergent that $\lim_{t\to\infty} \frac{\theta^t}{||\theta^t||_2} = u_{2,q}$, where $u_{2,q}$ is the maximum mixed-norm margin hyperplane of the training data. We further reveal natural interpretation of robustness that we obtain the maximum $\ell_2$-norm margin classifier under worst-case $\ell_q$-norm perturbation.

**Notations.** For two vectors $x, y \in \mathbb{R}^d$, $\langle x, y \rangle = \sum_{j=1}^{d} x_j y_j$ denotes their Euclidean inner product. For a vector $\theta \in \mathbb{R}^d$, $||\theta||_p$ defined by $||\theta||_p^p = \sum_{j=1}^{d} |\theta_j|^p$ denotes its $p$-norm for $p \in [1, \infty)$, and $||\theta||_\infty = \max_{j\in[d]} |\theta_j|$, where $[d] = \{1, \ldots, d\}$. For any general norm $|| \cdot ||$, we denote its dual norm by $||x||_* = \max_{||y||\leq 1} \langle x, y \rangle$. The sign function is $\text{sign}(v) = \mathbb{1}_{(v\geq 0)} - \mathbb{1}_{(v<0)}$. For a linear subspace $L \in \mathbb{R}^d$, we denote its orthogonal subspace by $L^\perp$.

## 2 BACKGROUND

We consider a binary classification problem using a dataset $\mathcal{S} = \{(x_i, y_i)\}_{i=1}^n \subset \mathbb{R}^d \times \{-1, +1\}$. We aim to learn a linear decision boundary $f(x) = \langle \theta, x \rangle$ and its associated classifier $\widehat{y}(x) = \text{sign}(f(x))$, by solving the empirical risk minimization problem:

$$\min_{\theta \in \mathbb{R}^d} \mathcal{L}(\theta; \mathcal{S}) = \min_{\theta \in \mathbb{R}^d} \sum_{i=1}^{n} \ell(y_i x_i^\top \theta), \text{ where } \ell(\cdot) \text{ is some loss function.} \tag{3}$$

In what follows, we suppress the explicit presentation of $\mathcal{S}$ when the context is clear, and we focus on the exponential loss $\ell(r) = \exp(-r)$. We point out that our analysis can be further extended to other smooth loss functions with tight exponential tail such as logistic loss.

We assume the dataset $\mathcal{S}$ is linearly separable, i.e., there exists $\overline{u}$ such that $\min_{i\in[n]} y_i x_i^\top \overline{u} > 0$. Under this assumption, one notable feature of problem (3) is that there is no finite minimizer, and $\mathcal{L}(\theta) \to 0$ only if $||\theta||_2 \to \infty$ along certain directions. In fact, there is a polyhedral cone $\mathcal{C}$, such that for any $u \in \mathcal{C}$, we have $\lim_{a\to\infty} \mathcal{L}(a\overline{u}) = 0$.

Several recent results have studied the implicit bias of gradient descent algorithm on separable dataset. Soudry et al. (2018) study the implicit bias of the gradient descent algorithm (GD) on (3), and show that $\lim_{t\to\infty} ||\theta^t||_2 = \infty$, while $\theta^t$ converges in direction to the maximum $\ell_2$-norm margin classifier (i.e., the standard SVM). Ji and Telgarsky (2018) further study the convergence of risk and parameter without separability condition. (Ji and Telgarsky, 2019) and (Gunasekar et al., 2018b) study the implicit bias for training deep linear network and linear convolutional networks, respectively. Gunasekar et al. (2018a) also analyze the implicit bias of steepest descent in general norm $|| \cdot ||$, and show that $\theta^t$ converges in direction to the maximum $|| \cdot ||_*$-norm margin hyperplane.

Throughout this paper, we assume the perturbation set is an $\ell_q$-norm ball with radius $c$, i.e., $\Delta = \{\delta \in \mathbb{R}^d : ||\delta||_q \leqslant c\}$. Under the general framework of adversarial training in (2), we aim to minimize the empirical adversarial risk

$$\min_{\theta \in \mathbb{R}^d} \mathcal{L}_{\text{adv}}(\theta) = \min_{\theta \in \mathbb{R}^d} \frac{1}{n} \sum_{i=1}^{n} \max_{\delta_i \in \Delta} \exp\left(-y_i (x_i + \delta_i)^\top \theta\right). \tag{4}$$

Note that, given any $\theta$, the inner maximization problem in (4) admits a closed form solution. Then the gradient descent based adversarial training (GDAT) algorithm runs iteratively that at the $t$-th iteration, we first solve the inner maximization problem by deriving the worst adversarial perturbation of each sample. It is not difficult to see that for each sample, the worst perturbation is $\delta_i^t = c y_i \delta_t^*$, where $\delta_t^* = \text{argmin}_{\delta:||\delta||_q \leq 1} \langle \delta, \theta^t \rangle$. Then, letting each sample's perturbed counterpart be $(\widetilde{x}_i^t, y_i) = (x_i + \delta_i^t, y_i)$, we take gradient of the loss function evaluated at the perturbed samples and perform a gradient descent step, i.e., $\theta^{t+1} = \theta^t - \eta^t \nabla_\theta \mathcal{L}\left(\theta^t; \{(\widetilde{x}_i^t, y_i)\}_{i=1}^n\right)$, where $\eta^t > 0$ is some prespecified stepsize. We present the outline of GDAT in Algorithm 1.

## 3 THEORETICAL RESULTS

In this section, we show that the GDAT algorithm possesses implicit bias, which depends on the perturbation set during training. We provide explicit characterization of the implicit bias, and further conclude that such implicit bias indeed promotes robustness against adversarial perturbation.

Let us start with some definitions. Consider a dataset $\mathcal{S} = \{(x_i, y_i)\}_{i=1}^n \subset \mathbb{R}^d \times \{-1, +1\}$. Given $p, q > 0$ such that $1/p + 1/q = 1$, the $\ell_q$-norm margin of $H_\theta$ on $\mathcal{S}$ is defined as $\gamma_q(\theta) = \min_{i \in [n]} y_i x_i^\top \theta / ||\theta||_p$. Note that for $x_i \in \mathbb{R}^d$, $|\theta^\top x| / ||\theta||_p$ measures the $\ell_q$ distance between $x_i$ and the hyperplane $H_\theta = \{x \in \mathbb{R}^d : \theta^\top x = 0\}$. Since $y_i \in \{-1, +1\}$, when $H_\theta$ correctly classifies all samples, $\gamma_q(\theta)$ measures the minimal $\ell_q$ distance between the samples in $\mathcal{S}$ and $H_\theta$. Given that $\gamma_q(\theta)$ is scale-invariant with respect to $\theta$, without loss of generality, we restrict $||\theta||_p = 1$. We also identify the hyperplane $H_\theta$ by its normal vector $\theta$.

---

**Algorithm 1** Gradient Descent based Adversarial Training (GDAT) with $\ell_q$-norm Perturbation

---

**Input:** Number of iterations $T$, perturbation level $c$, stepsizes $\{\eta^t\}_{t=0}^T$, samples $\{x_i, y_i\}_{i=1}^n$.
**Initialize:** $\theta^0 \leftarrow 0$.
**for** $t = 0, \ldots, T-1$ **do**
    **for** $i = 1, \ldots, n$ **do**
        Compute $\delta_i^t = c y_i \operatorname{argmin}_{||\delta||_q \leq 1} \langle \delta, \theta^t \rangle$
        Let $(\widetilde{x}_i^t, y_i) \leftarrow (x_i + \delta_i^t, y_i)$.
    **end for**
    $\theta^{t+1} \leftarrow \theta^t - \frac{\eta^t}{n} \sum_{i=1}^n \exp\left(-y_i \widetilde{x}_i^\top \theta^t\right)(-y_i \widetilde{x}_i)$.
**end for**

---

**Definition 3.1.** *For $p, q > 0$ with $1/p + 1/q = 1$, the maximum $\ell_q$-norm margin hyperplane $u_q$ of $\mathcal{S} = \{(x_i, y_i)\}_{i=1}^n \subset \mathbb{R}^d \times \{-1, +1\}$ and its associated $\ell_q$-norm margin $\gamma_q$ are defined as*

$$u_q \in \operatorname*{argmax}_{||\theta||_p = 1} \min_{i \in [n]} y_i x_i^\top \theta, \quad \gamma_q = \max_{||\theta||_p = 1} \min_{i \in [n]} y_i x_i^\top \theta. \tag{5}$$

*We denote* $\mathrm{SV}(\mathcal{S})$ *as the support vectors of $\mathcal{S}$, i.e.,* $\mathrm{SV}(\mathcal{S}) = \operatorname{argmin}_{(x,y) \in \mathcal{S}} \langle u_q, yx \rangle$.

By the separability assumption, $u_q$ is an optimal hyperplane that correctly classifies all samples with the maximal margin $\gamma_q > 0$. Next, by the notion of margin defined above, we characterize the landscape of empirical adversarial risk in (4) based on the perturbation level $c$.

**Proposition 3.1.** *Let $p, q > 0$ satisfy $1/p + 1/q = 1$. Given a nonnegative scalar $c$, where $0 \leq c < \gamma_q = \max_{||\theta||_p \leq 1} \min_{i \in [n]} y_i x_i^\top \theta$, problem (4) has infimum 0 but does not admit a finite minimizer. When $c > \gamma_q$, problem (4) has a unique finite minimizer $\widehat{\theta}(c)$, and is equivalent to the standard clean training with explicit $\ell_p$-norm regularization. That is, there exists $\lambda(c) > 0$ such that*

$$\widehat{\theta}(c) = \operatorname*{argmax}_{\theta \in \mathbb{R}^d} \frac{1}{n} \sum_{i=1}^n \exp(-y_i x_i^\top \theta) + \lambda(c) ||\theta||_p.$$

It is not difficult to see that for $c < \gamma_q$, any perturbed dataset $\widetilde{\mathcal{S}} = \{(\widetilde{x}_i, y_i)\}_{i=1}^n$, with $||x_i - \widetilde{x}_i||_q \leqslant c$ for all $i$, is still linearly separable, which directly follows from the definition of $\gamma_q$ above. On the other hand, when $c > \gamma_q$, by the definition of $\gamma_q$, there exists some perturbed dataset $\widetilde{\mathcal{S}} = \{(\widetilde{x}_i, y_i)\}_{i=1}^n$, with $||x_i - \widetilde{x}_i||_q \leqslant c$ for all $i$, such that $\widetilde{\mathcal{S}}$ is no longer linearly separable.

### 3.1 Adversarial Perturbation with Bounded $\ell_2$-Norm

In this subsection, we analyze both the empirical adversarial risk convergence and the parameter convergence of the case when the perturbation set $\Delta$ in (4) is an $\ell_2$-norm ball with radius $c$.

**Adversarial Risk Convergence.** We first analyze the convergence of empirical adversarial risk (4) using GDAT. One substantial roadblock of minimizing (4) is its non-smoothness, in the sense that $\mathcal{L}_{\mathrm{adv}}(\theta)$ is not differentiable at the origin, and its Hessian $\nabla^2 \mathcal{L}_{\mathrm{adv}}(\theta)$ explodes around the origin. To address the challenge, our key observation is that, by the next lemma, at each iteration, there exists an acute angle between the update on $\theta^t$ and the maximum $\ell_2$-norm margin hyperplane $u_2$. This gives a lower bound on $||\theta^t||_2$.

**Lemma 3.1.** *Take $\Delta = \{\delta \in \mathbb{R}^d : ||\delta||_2 \leq c\}$ in problem (4). Given $c < \gamma_2$, we have that $\langle -\nabla \mathcal{L}_{\mathrm{adv}}(\theta), u_2 \rangle \geq \mathcal{L}_{\mathrm{adv}}(\theta)(\gamma_2 - c) > 0$ for any $\theta \in \mathbb{R}^d$.*

We highlight that despite its simple proof, Lemma 3.1 and its generalization to $\ell_q$-perturbation is a crucial step for analyzing both adversarial risk and implicit bias. In addition, our techniques here can also be adapted to simplify the proof of Lemma 10 in (Gunasekar et al., 2018a), which, in comparison, is more technically involved.

Since we initialize GDAT (Alg. 1) using $\theta^0 = 0$, any perturbation inside $\Delta$ will have no effect on the adversarial loss. Hence we take clean samples as adversarial examples at the first iteration of

GDAT. From Lemma 3.1, we have the following simple corollary showing that our whole solution path $\{\theta^t\}_{t=1}^T$ is bounded away from the origin.

**Corollary 3.1.** *Let $\theta^0 = 0$ in Algorithm 1 with $q = 2$, we have:* $||\theta^t||_2 \geq \eta^0 \gamma_2$ *for all $t \geq 1$.*

By Corollary 3.1, we bypass the non-differentiability issue at the origin and also control the Hessian $\nabla^2 \mathcal{L}_{\mathrm{adv}}(\theta)$ throughout the entire training process. Similar to (Ji and Telgarsky, 2018), in the next theorem, we show that the loss $\mathcal{L}_{\mathrm{adv}}(\theta)$, although not uniformly smooth, is locally $\mathcal{L}_{\mathrm{adv}}(\theta)$-smooth. Consequently, by the smoothness based analysis of the gradient descent algorithm, we establish the convergence of the empirical adversarial risk.

**Theorem 3.1.** *Suppose $||x_i||_2 \leq 1$ for all $i = 1 \ldots n$. For GDAT (Alg. 1) with $\ell_2$-norm perturbation, i.e., $\Delta = \{\delta \in \mathbb{R}^d : ||\delta||_2 \leqslant c\}$, we set $c < \gamma_2$, $\eta^0 = 1$ and $\eta^t = \eta \leq \min\{\frac{\gamma_2/e}{(1+c)^3 \gamma_2 + 2c(1+c)}, 1\}$ for $t \geq 1$, then we have*

$$\frac{1}{n} \sum_{i=1}^n \max_{\delta_i \in \Delta} \exp\left(-y_i(x_i + \delta_i)^\top \theta^t\right) = \mathcal{O}\left(\frac{\log^2 t}{t\eta(\gamma_2 - c)^2}\right). \tag{6}$$

In comparison with the standard clean training using GD (Ji and Telgarsky, 2018), this theorem states that we pay an extra $(\gamma_2 - c)^{-2}$ factor in the risk convergence of adversarial training. However, this direct comparison is too pessimistic since we compare the adversarial risk with the standard risk (corresponding to $\Delta = \{0\}$). Interestingly, as seen later in Corollary 3.2, we prove that the convergence of standard risk in GDAT is significantly faster than its counterpart in the standard clean training using GD.

**Parameter Convergence.** We then show that if we set the perturbation level $c < \gamma_2$ in the GDAT algorithm, GDAT with $\ell_2$-norm perturbation possesses the same implicit bias as the standard clean training using GD, i.e., we have $\lim_{t\to\infty} \frac{\theta^t}{||\theta^t||_2} = u_2$. Intuitively, GDAT with $\ell_2$-norm perturbation searches for a decision hyperplane that is robust to $\ell_2$-norm perturbation. Since the learned decision hyperplane in the standard clean using GD converges to $u_2$, which is already the most robust decision hyperplane against $\ell_2$-norm perturbation to the data, GDAT retains the implicit bias of standard clean training using GD.

Surprisingly, even though both GDAT in the adversarial training and GD in the standard clean training converge in directions to $u_2$, their rates of directional convergence are significantly different as shown later. Specifically, letting the perturbation level $c$ depend on the total number of iterations $T$ in the GDAT algorithm, the directional error after $T$ iterations in GDAT algorithm can be significantly smaller than the error of GD in the standard clean training.

We first show that the projection of $\theta^t$ onto the orthogonal subspace of $\mathrm{span}(u_2)$ is bounded.

**Lemma 3.2.** *Define $\alpha(\mathcal{S}) = \min_{||\xi||_2=1, \xi \in \mathrm{span}(u_2)^\perp} \max_{(x,y) \in \mathrm{SV}(\mathcal{S})} \langle \xi, yx \rangle$, where we assume $\mathrm{SV}(\mathcal{S})$ spans $\mathbb{R}^d$. Let $\theta_\perp$ be the projection of vector $\theta$ onto $\mathrm{span}(u_2)^\perp$. Then there exists a constant $K$ that only depends on $\alpha(\mathcal{S})$ and $\log n$, such that $||\theta_\perp^t||_2 \leq K$ for any $t \geq 0$ in the GDAT algorithm.*

Note that the same $\alpha(\mathcal{S})$ is defined in (Ji and Telgarsky, 2019) and proved to be positive with probability 1 if the data is sampled from absolutely continuous distribution. We then show in the next lemma that $||\theta^t||_2$ goes to infinity, where we provide a refined analysis to establish the acceleration of the directional convergence in comparison with the standard clean training.

**Lemma 3.3.** *Under the same conditions in Theorem 3.1, and let $\alpha = \alpha(\mathcal{S})$ defined in Lemma 3.2. Then for all $t \geq 0$, we have*

$$||\theta^t||_2 \geq \log\left(\frac{t\eta(\gamma_2 - c)^2}{n^{1+1/\alpha} \log^2 t}\right)/(\gamma_2 - c).$$

Lemma 3.3 provides the key insight to establish the acceleration of directional convergence. Specifically, it allows us to set $c$ depending on the total number of iterations $T$, so that $||\theta^T||_2$ is sublinear in $T$, in comparison with being logarithmic in $T$ in standard clean training as in Ji and Telgarsky (2018). We are now ready to present the main theorem for parameter convergence.

**Theorem 3.2** (Speed-up of Parameter Convergence). *Under same conditions in Theorem 3.1, and let $\alpha = \alpha(\mathcal{S})$ and $K$ be defined in Lemma 3.2. In GDAT with $\ell_2$-norm perturbation, let $c$ and total*

*number of iterations $T$ satisfy $\gamma_2 - c = \left(\frac{n^{1+1/\alpha} \log T}{\eta T}\right)^{1/2}$, and define $\overline{\theta}^T = \frac{\theta^T}{||\theta^T||_2}$. We have*

$$1 - \left\langle \overline{\theta}^T, u_2 \right\rangle = \mathcal{O}\left(\frac{n^{(1+1/\alpha)/2} K \log T}{\sqrt{\eta}\sqrt{T}}\right). \tag{7}$$

One might argue that the polynomial dependence on sample size $n$ in (7) is too pessimistic, making the GDAT unfavorable in comparison with the standard clean training. We show that this is not an issue by a direct comparision of iteration complexity to achieve $||\overline{\theta}^T - u_2||_2 \le \epsilon$ for a given precision $\epsilon > 0$. Specifically, given $\epsilon > 0$, to achieve $||\overline{\theta}^T - u_2||_2 \le \epsilon$, GDAT needs $\widetilde{\mathcal{O}}\left(n^{(1+1/\alpha)}\epsilon^{-2}\right)$ number of iterations. In comparison, the standard clean training by GD needs $\widetilde{\mathcal{O}}\left(n \exp\left(\epsilon^{-1}\right)\right)$ number of iterations (Ji and Telgarsky, 2018), which has exponential dependence on precision $\epsilon$.

Finally, by Theorem 3.1 and Lemma 3.3, we show that the empirical clean risk after $T$ iterations of GDAT is almost exponentially smaller than its counterpart in the standard clean training.

**Corollary 3.2** (Speed-up of Clean Risk Convergence)**.** *Under the same conditions in Theorem 3.2, we have*

$$\mathcal{L}(\theta^T) = \mathcal{O}\left(\exp\left(-\mu\sqrt{T}/\log T\right)\right),$$

*where $\mu$ is a constant dependent on $\eta, \alpha, n$.*

Note that the empirical clean risk decreases at the rate of $\mathcal{O}\left(\exp(-\sqrt{T})\right)$ up to a logarithmic factor in the exponent. In comparison, using standard clean training with GD, we only have $\mathcal{L}(\theta^T) = \mathcal{O}\left(1/T\right)$ (Soudry et al., 2018).

## 3.2 ADVERSARIAL PERTURBATION WITH BOUNDED $\ell_q$-NORM

In this subsection, we generalize our results to the case where the perturbation set is some bounded $\ell_q$-norm ball. To facilitate our discussion, we first define a robust version of SVM.

**Definition 3.2.** *For a given separable dataset $\mathcal{S}$ with $\ell_q$-norm margin $\gamma_q$ and $c < \gamma_q$, letting $1/p + 1/q = 1$, the robust SVM against $\ell_q$-norm perturbation parameterized by $c$ is*

$$\min_{\theta \in \mathbb{R}^d} \frac{1}{2}||\theta||_2^2 \quad s.t. \quad y_i x_i^\top \theta \ge c||\theta||_p + 1, \forall i = 1, \ldots, n. \tag{8}$$

**Remark 3.1** (Maximum Mixed-norm Margin)**.** *Note that problem (8) is equivalent to solving for a maximum mixed-norm margin hyperplane. Specifically, by the KKT condition of (8), there exists $\eta(c) > 0$, such that (8) is equivalent to the following problem:*

$$\min_{\theta \in \mathbb{R}^d} ||\theta||_2 + \eta(c)||\theta||_p \quad s.t. \quad y_i x_i^\top \theta \ge 1, \forall i = 1, \ldots, n. \tag{9}$$

*Now define $|| \cdot || = || \cdot ||_2 + \eta(c)|| \cdot ||_p$, it is clear that $|| \cdot ||$ defines a norm which is a mixture of $\ell_2$ and $\ell_p$ norm. Let $|| \cdot ||_*$ be its dual norm. Then we have that the solution to (9) is the maximum $|| \cdot ||_*$-norm margin hyperplane.*

Note that the constraint in (8) is equivalent to $\min_{||\delta_i||_q \le c} y_i(x_i + \delta_i)^\top \theta \ge 1, \forall i = 1, \ldots, n$. By a simple scaling argument, in the following lemma, we see the robust nature of (8).

**Lemma 3.4.** *Under the same notations in Definition 3.2, problem (8) is equivalent to:*

$$\gamma_{2,q}(c) = \max_{||\theta||_2=1} \min_{i\in[n]} \min_{||\delta_i||_q \leqslant c} y_i(x_i + \delta_i)^\top \theta. \tag{10}$$

We denote the (unique) solution to problem (10) as $u_{2,q}(c)$. In what follows, we surpress explicit presentation of $c$ when the context is clear.

The equivalent formulation (10) provides a clear interpretation on the robustness of (10). In particular, the robust SVM against $\ell_q$-norm perturbation parameterized by $c$ is in fact the SVM problem on the the dataset $\mathcal{S}(c, q)$, which is generated from $\mathcal{S}$ by placing a $\ell_q$-norm ball with radius $c$ around each samples, i.e., $\mathcal{S}(c, q) = \{(x, y) : \exists i \in [n], s.t., ||x - x_i||_p \leqslant c, y = y_i\}$. In other words, $u_{2,q}$ is the maximum $\ell_2$-norm margin classifier under worst case $\ell_q$-norm perturbation bounded by $c$.

In the remaining part of this section, we first analyze the convergence of the empirical adversarial risk, and then establish the implicit bias of GDAT with $\ell_q$ perturbation for $q \in [1, \infty]$. Our analysis

for $q \in \{1, \infty\}$ is based on approximation argument. For ease of presentation, we only discuss when $q \in (1, \infty)$ in the main text, and defer the discussion for $q \in \{1, \infty\}$ in Appendix D.

**Adversarial Risk Convergence.** Our analysis is similar to the analysis for GDAT with $\ell_2$ perturbation, where we use similar techniques to address issues such as non-differentiability at the origin and Hessian explosion of $\mathcal{L}_{\mathrm{adv}}(\theta)$ around the origin.

**Theorem 3.3.** *Suppose* $||x_i||_2 \leq 1$ *for* $i = 1, \ldots, n$, *and let* $\frac{1}{p} + \frac{1}{q} = 1$. *In the GDAT with* $\ell_q$-*norm perturbation, setting* $c < \gamma_q$ *and letting* $M_p = \left[ (1 + c\sqrt{d})^2 + \frac{c(p-1)}{\gamma_{2,q}} d^{\frac{3p-2}{2p-2}} \right] \exp\left( -\gamma_{2,q}^2 + c\sqrt{d} \right)$, *set* $\eta^0 = 1$ *and* $\eta^t = \eta \leq \min\{\frac{1}{M_p}, 1\}$ *for* $t \geq 1$. *We have that*

$$\frac{1}{n} \sum_{i=1}^{n} \max_{\delta_i \in \Delta} \exp\left( -y_i (x_i + \delta_i)^\top \theta^t \right) = \mathcal{O}\left( \frac{\log^2 t}{t\eta\gamma_{2,q}^2} \right). \tag{11}$$

We point out here that (6) is a special case of (11). In particular, by the definition of $\gamma_{2,q}(c)$, we have that $\gamma_{2,2}(c) = \gamma_2 - c$, which recovers bound (6) from (11).

**Parameter Convergence.** We show that if we set $c < \gamma_q$ in the GDAT algorithm with stepsizes specified in Theorem 3.3, with $\ell_q$ perturbation, the algorithm still possesses implicit bias property, i.e., $\theta^t$ still has directional convergence, and the limiting direction depends on the perturbation set $\Delta$.

**Theorem 3.4** (Implicit Bias of GDAT with $\ell_q$-norm Perturbation). *Under the same conditions in Theorem 3.3, define* $\overline{\theta}^t = \frac{\theta^t}{||\theta^t||_2}$, *then we have:*

$$1 - \left\langle \overline{\theta}^t, u_{2,q} \right\rangle = \mathcal{O}\left( \frac{\log n}{\log t} \right)$$

Combining Theorem 3.4 and Lemma 3.4, we conclude that GDAT with $\ell_q$-norm perturbation indeed promotes robustness against $\ell_q$ perturbation. Using GDAT with $\ell_q$-norm perturbation will result in a classifier which is the maximum $\ell_2$-norm margin classifier under worst case $\ell_q$-norm perturbations to the samples bounded by $c$. The learned classifier will have $\ell_q$-norm margin at least $c$. As we increase perturbation level $c$ to $\gamma_q$, the learned classifier will converge to maximum $\ell_q$-norm margin classifier.

# 4 NUMERICAL EXPERIMENT

In this section, we first conduct numerical experiments on linear classifiers to backup our theoretical findings. We further empirically extend our method to neural networks, where our numerical results demonstrate that our theoretical results can be potentially generalized.

**Linear Classifiers.** We investigate the empirical performance of the GDAT algorithm on linear classifiers, with training set $\mathcal{S} = \{((-0.5, 1), +1), ((-0.5, -1), -1), ((-0.75, -1), -1), ((2, 1), +1)\}$. It is straightforward to verify that the maximum $\ell_2$-norm margin classifier is $u_2 = (0, 1)$.

Considering $\ell_2$-norm perturbations, we first run standard clean training with GD, and GDAT with $\ell_2$-norm perturbation ($c = 0.95\gamma_2$), for $2.5 \times 10^4$ number of iterations. In both GD and GDAT we take constant stepsizes, with $\eta = 1$ and $\eta = 0.1$, respectively. By Figure 1(a), we see that the convergence rate of adversarial loss using GDAT is similar to the convergence rate of clean loss using GD. However, when we directly compare the clean losses of GDAT and GD, GDAT clearly demonstrates an exponential speed-up in comparison with GD, which is consistent with Corollary 3.2. Additionally, as pointed out by Theorem 3.2, GDAT also enjoys significant speed-up in terms of the directional convergence of $\theta^t$ to $u_2$. We also compare the norm growth $||\theta^t||_2$, and observe that the norm generated by GDAT grows much faster than the norm generated by GD, which is also in alignment with our discussions in Section 3.1.

We further run GDAT with $\ell_\infty$-norm perturbation ($c = 0.5$). By Lemma 3.4, we have that $u_{2,\infty} = (0, 1)$. Note that the Hausdorff distance between $\ell_q$-norm ball and $\ell_\infty$-norm ball distance goes to zero as $q$ goes to infinity. Thus, we have that (10) for $q = 1000$ is a close approximation of (10) for $q = \infty$. We run two versions of GDAT, where one uses $\ell_q$-norm perturbation with $q = 1000$, and the other uses $\ell_\infty$-norm perturbation. We run both algorithms with stepsize $\eta = 0.1$ for $5.0 \times 10^5$ number of iterations, and we present the results in Figure 1(b). We find that the two training methods behave similarly. In addition, the empirical directional convergence rates of $\theta^t$ just differ slightly.

**Neural Networks.** It is seen above that GDAT with $\ell_2$-norm perturbations converges significantly faster than GD for linear classifiers in adversarial training. A natural question is whether this is still the

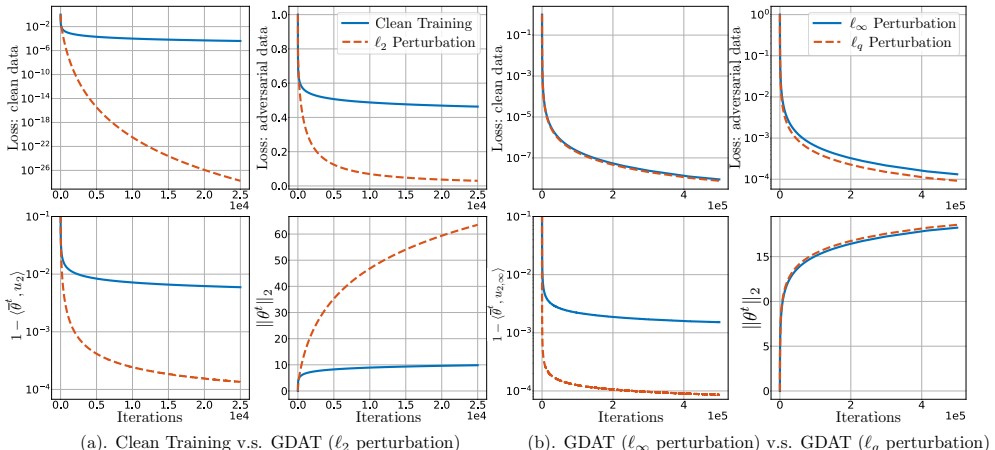

(a). Clean Training v.s. GDAT ($\ell_2$ perturbation)  (b). GDAT ($\ell_\infty$ perturbation) v.s. GDAT ($\ell_q$ perturbation)

Figure 1: GDAT of Linear Classifiers.

case on adversarial training of more complicated neural networks. We conduct experiments on neural network with one hidden layer. We take the two classes from MNIST dataset with label "2" and "9" to form our training set $\mathcal{S}$. We also vary the width of the hidden layer in $\{64 \times 64, 128 \times 128, 256 \times 256\}$.

One major difference from the case of linear classifiers is that we cannot solve the inner maximization problem of (2) exactly as it does not admits a closed-form solution. Instead, we solve the inner problem approximately using projected gradient descent with 20 iterations and stepsize 0.01. We test two versions of GDAT, where one adopts $\ell_2$-norm perturbations ($c = 2.8$), and the other uses $\ell_\infty$-norm perturbations ($c = 0.1$). For standard clean training and the outer minimization problem in (2), we use the stochastic gradient descent algorithm with batch size 128 and constant stepsize $10^{-5}$.

We compare the loss and classification accuracy, which are evaluated using the clean training samples, of standard clean training and GDAT. By Figure 2, we see that GDAT indeed accelerates the convergence of both loss and classification accuracy on clean training samples. The performance gap is most obvious when the width of the hidden layer is small, and reduces gradually as we increase the width of the hidden layer. We argue that such reduction comes from the fact that as network width increases, the margin on the samples outputted by the hidden layer also increases. As suggested by Theorem 3.2, in this case, a larger perturbation level $c$ should be used. We conduct additional experiments with various perturbation level in Appendix E to empirically verify our argument.

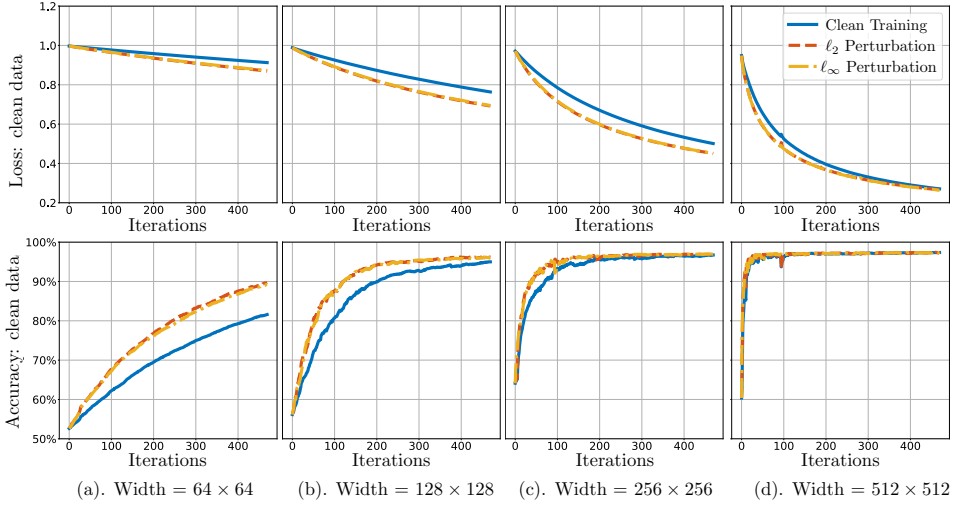

(a). Width $= 64 \times 64$    (b). Width $= 128 \times 128$    (c). Width $= 256 \times 256$    (d). Width $= 512 \times 512$

Figure 2: GDAT of Neural Network on MNIST Dataset.

## 5 DISCUSSIONS

We investigate the implicit bias of GDAT for linear classifier. There are several plausible natural extensions. For example, we can represent a linear classifier using a **deep linear network**, which is significantly overparameterized. Some recent results characterize the implicit bias of gradient descent for training deep linear networks (Ji and Telgarsky, 2019) and linear convolutional networks (Gunasekar et al., 2018b). Motivated by these results, investigating the implicit bias of GDAT in training deep linear networks worths future investigations.

Meanwhile, investigating implicit bias in **deep nonlinear networks** is a more important and challenging direction: (1) For linear classifiers, adding adversarial perturbations during training can be understood as a form of regularization, which explains the faster convergence in training. Although observed empirically, the potential acceleration of adversarial training is not yet understood in the current literature, to the best of our knowledge. (2) The notion of margin for neural networks still lacks proper definition, which we need to define to facilitate investigations on the effect of adversarial training in promoting robustness. (3) Ultrawide nonlinear networks have been shown to evolve similarly to linear networks using gradient descent (Ghorbani et al., 2019; Lee et al., 2019). We shall further investigate if our results on linear classifiers can be extended to wide nonlinear networks.

## 6 ACKNOWLEDGEMENTS

Fang is partially supported by NSF DMS-1820702 and NSF DMS-1953196.

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

## A   PROOF OF PROPOSITION 3.1

*Proof.* Suppose $c < \gamma_q$. Letting $\theta_\alpha = \alpha u_q$ for $\alpha > 0$, we have

$$\mathcal{L}_{\mathrm{adv}}(\theta_\alpha) = \frac{1}{n} \sum_{i=1}^{n} \exp\left(-y_i x_i^\top \theta_\alpha + c||\theta_\alpha||_p\right)$$

$$= \frac{1}{n} \sum_{i=1}^{n} \exp\left(-\alpha y_i x_i^\top u_q + c\alpha\right)$$

$$\leq \frac{1}{n} \sum_{i=1}^{n} \exp\left(-\alpha \gamma_q + c\alpha\right).$$

Letting $\alpha \to \infty$, we obtain $\lim_{\alpha \to \infty} \mathcal{L}_{\mathrm{adv}}(\theta_\alpha) = 0$, which implies $\inf_{\theta \in \mathbb{R}^d} \mathcal{L}_{\mathrm{adv}}(\theta) = 0$. Note that $\mathcal{L}(\theta)$ does not admit any finite minimizer since $\mathcal{L}_{\mathrm{adv}}(\theta) > 0$ for any $\theta \in \mathbb{R}^d$.

If $c > \gamma_q$, by the definition of maximum $\ell_q$-norm margin, for any $\theta \in \mathbb{R}^d$, there exists $(y_i, x_i) \in \mathcal{S}$ for some $i \in [n]$ such that $y_i x_i^\top \theta \leqslant \gamma_q ||\theta||_p$. Hence, $\mathcal{L}_{\mathrm{adv}}(\theta) \geqslant \exp\left(n^{-1}(c - \gamma_q)||\theta||_p\right)$. Then it is easy to see that $\mathcal{L}_{\mathrm{adv}}(\theta)$ has bounded sublevel set and hence a finite minimizer $\widehat{\theta}$. Since $\mathcal{L}_{\mathrm{adv}}(\theta)$ is convex, we examine its first-order KKT condition, given by

$$\frac{1}{n} \sum_{i=1}^{n} \exp\left(-y_i x_i^\top \widehat{\theta} + c||\widehat{\theta}||_p\right) \left(-y_i x_i + c\partial ||\widehat{\theta}||_p\right) \ni 0. \tag{12}$$

Consider the regularized problem with regularization parameter $\eta$:

$$\min_{\theta \in \mathbb{R}^d} \frac{1}{n} \sum_{i=1}^{n} \exp\left(-y_i x_i^\top \theta\right) + \eta ||\theta||_p.$$

Its first-order KKT condition is

$$\frac{1}{n} \sum_{i=1}^{n} \exp\left(-y_i x_i^\top \theta\right) \left(-y_i x_i\right) + \eta \partial ||\theta||_p \ni 0. \tag{13}$$

Looking at (12) and (13) together, by taking $\eta = \frac{c}{n} \sum_{i=1}^{n} \exp\left(-y_i x_i^\top \widehat{\theta} + c||\widehat{\theta}||_p\right)$, we have that the solution to the adversarial training problem $\widehat{\theta}$ is also the solution to the regularized problem. $\square$

To facilitate our later discussions, we point out that by the conjugacy of $\ell_p$-norm and $\ell_q$-norm, (4) has the following equivalent form that

$$\min_{\theta \in \mathbb{R}^d} \mathcal{L}_{\mathrm{adv}}(\theta) = \min_{\theta \in \mathbb{R}^d} \frac{1}{n} \sum_{i=1}^{n} \exp\left(-y_i x_i^\top \theta + c||\theta||_p\right). \tag{14}$$

In fact, one can verify that the GDAT algorithm is equivalent to gradient descent algorithm on (14).

## B  PROOFS FOR SECTION 3.1

*Proof of Lemma 3.1.* Recall we have $\mathcal{L}_{\text{adv}}(\theta) = \frac{1}{n} \sum_{i=1}^{n} \max_{||\delta||_2 \leq c} \exp\left(-y_i(x_i + \delta_i)^\top \theta\right)$. For each sample $(x_i, y_i) \in \mathcal{S}$, given a classifier $\theta$, the worse case perturbation is $\widetilde{\delta}_i = \operatorname{argmax}_{||\delta||_2 \leq c} \exp\left(-y_i(x_i + \delta)^\top \theta\right) = \operatorname{argmin}_{||\delta||_2 \leq c} y_i \delta^\top \theta$. The corresponding loss is $\mathcal{L}_{\text{adv}}(\theta) = \frac{1}{n} \sum_{i=1}^{n} \exp\left(-y_i(x_i + \widetilde{\delta}_i)^\top \theta\right)$.

Since for a fixed $\delta_i$, the function $\exp\left(-y_i(x_i + \delta_i)^\top \theta\right)$ is convex in $\theta$, hence the gradient of $\mathcal{L}_{\text{adv}}(\theta)$ is

$$-\nabla \mathcal{L}_{\text{adv}}(\theta) = \frac{1}{n} \sum_{i=1}^{n} \exp\left(-y_i(x_i + \widetilde{\delta}_i)^\top \theta\right) y_i(x_i + \widetilde{\delta}_i).$$

Then from the definition of $u_2$ (5), we have

$$\left\langle -\nabla \mathcal{L}_{\text{adv}}(\theta), u_2 \right\rangle = \sum_{i=1}^{n} \exp\left(-y_i(x_i + \widetilde{\delta}_i)^\top \theta\right) \left\langle y_i(x_i + \widetilde{\delta}_i), u_2 \right\rangle \tag{15}$$

$$\geq \sum_{i=1}^{n} \exp\left(-y_i(x_i + \widetilde{\delta}_i)^\top \theta\right) \left(\langle y_i x_i, u_2 \rangle - c\right) \tag{16}$$

$$\geq \sum_{i=1}^{n} \exp\left(-y_i(x_i + \widetilde{\delta}_i)^\top \theta\right) (\gamma_2 - c) = \mathcal{L}_{\text{adv}}(\theta)(\gamma_2 - c), \tag{17}$$

where in the second inequality holds since $||\widetilde{\delta}_i||_2 \leqslant c$ and $||u_2||_2 = 1$. $\qquad\square$

*Proof of Corollary C.1.* Since $\mathcal{L}_{\text{adv}}(\theta)$ is not differentiable at $\theta^0 = 0$, we use subgradient (note that $\mathcal{L}_{\text{adv}}(\theta)$ is convex) at 0. Specifically, we take $\nabla \mathcal{L}_{\text{adv}}(\theta^0) = \frac{1}{n} \sum_{i=1}^{n} z_i \in \partial \mathcal{L}_{\text{adv}}(\theta^0)$. Then we have $\langle \theta^1, u_2 \rangle = \frac{\eta^0}{n} \sum_i \langle z_i, u_2 \rangle \geq \eta^0 \gamma_2$, where the last inequality uses the definition of $\gamma_2$.

By Lemma 3.1, we have $\langle \theta^t, u_2 \rangle \geq \eta^0 \gamma_2$ for all $t \geq 1$, which also implies $\langle v, u_2 \rangle \geq \eta^0 \gamma_2$ and hence $||v^t||_2 \geq \eta^0 \gamma_2$ for $v \in \left[\theta^t, \theta^{t+1}\right]$. $\qquad\square$

*Proof of Theorem 3.1.* For simplicity, we let $z_i = y_i x_i$, where we have $||z_i||_2 \leq 1$ as we assume $||x_i||_2 \leq 1$. We have

$$\nabla \mathcal{L}_{\text{adv}}(\theta) = \frac{1}{n} \sum_{i=1}^{n} \exp\left(-z_i^\top \theta + c||\theta||_2\right) \left(-z_i + c\frac{\theta}{||\theta||_2}\right),$$

$$\nabla^2 \mathcal{L}_{\text{adv}}(\theta) = \frac{1}{n} \sum_{i=1}^{n} \exp\left(-z_i^\top \theta + c||\theta||_2\right) \left(-z_i + c\frac{\theta}{||\theta||_2}\right) \left(-z_i + c\frac{\theta}{||\theta||_2}\right)^\top$$

$$+ \frac{1}{n} \sum_{i=1}^{n} \exp\left(-z_i^\top \theta + c||\theta||_2\right) c \left(||\theta||I - \frac{\theta\theta^\top}{||\theta||_2}\right) / ||\theta||_2^2$$

$$= \frac{1}{n} \sum_{i=1}^{n} \exp\left(-z_i^\top \theta + c||\theta||_2\right) \left[z_i z_i^\top - 2\frac{c z_i^\top \theta}{||\theta||_2} + c^2 \theta\theta^\top / ||\theta||_2^2 + cI/||\theta||_2 - c\theta\theta^\top / ||\theta||_2^3\right].$$

Note that the Hessian expression indicates that the objective is highly non-smooth around origin, and the loss is not even differentiable at origin. However, we shall prove that starting from origin, every iteration generated by GADT stays away from the origin with distance bounded below.

Using Taylor's expansion, and by definition $\theta^{t+1} = \theta^t - \eta^t \nabla \mathcal{L}_{\text{adv}}(\theta^t)$, we have

$$\mathcal{L}_{\text{adv}}(\theta^{t+1}) \leq \mathcal{L}_{\text{adv}}(\theta^t) - \eta^t ||\nabla \mathcal{L}_{\text{adv}}(\theta^t)||_2^2 + \frac{(\eta^t ||\nabla \mathcal{L}_{\text{adv}}(\theta^t)||)^2}{2} \max_{v \in [\theta^t, \theta^{t+1}]} \lambda(H(v))_{\max}, \tag{18}$$

where $\lambda(H(v))_{\max}$ denotes the largest eigenvalue of $H(v)$, where

$$H(v) = \frac{1}{n} \sum_{i=1}^{I} n \exp\left(-z_i^\top v + c||v||_2\right) \left[z_i z_i^\top - 2\frac{c z_i^\top v}{||v||_2} + c^2 vv^\top / ||v||_2^2 + cI/||v||_2 - cvv^\top / ||v||_2^3\right].$$

To upper bound $H(v)$, we need a lower bound on $||v||$, which is readily given by Corollary C.1. That is, $||v||_2 \geq \eta^0 \gamma_2$.

We now analyze (18) for $t \geq 1$, where we show that $\mathcal{L}_{\mathrm{adv}}(\theta^t)$ is locally smooth with parameter proportional to $\mathcal{L}_{\mathrm{adv}}(\theta^t)$, and with proper stepsize, the risk is monotonely decreasing. Note that $z_i z_i^t \leq I$, $-2c\frac{z_i^\top v}{||v||_2} \leq 2cI$, $c^2 v v^\top/||v||_2^2 \leq c^2 I$. Now since $||v^t||_2 \geq \eta^0 \gamma_2$, we have $cI/||v||_2 - cvv^\top/||v||_2^3 \leq \frac{2c}{\eta^0\gamma_2}I$. Plugging them in, we have

$$H(v) \leq \frac{1}{n}\sum_i \exp\left(-z_i^\top v + c||v||_2\right)\left(1 + 2c + c^2 + \frac{2c}{\eta^0\gamma_2}\right)I$$

$$= L_{adv}(v)\left(1 + 2c + c^2 + \frac{2c}{\eta^0\gamma_2}\right)I,$$

and (18) reduces to

$$\begin{aligned}
\mathcal{L}_{\mathrm{adv}}(\theta^{t+1}) \leq &\mathcal{L}_{\mathrm{adv}}(\theta^t) - \eta^t||\nabla\mathcal{L}_{\mathrm{adv}}(\theta^t)||_2^2 \\
&+ \frac{(\eta^t||\nabla\mathcal{L}_{\mathrm{adv}}(\theta^t)||)^2}{2}\left[(1+c)^2 + \frac{2c}{\eta^0\gamma_2}\right]\max\left\{\mathcal{L}_{\mathrm{adv}}(\theta^t), \mathcal{L}_{\mathrm{adv}}(\theta^{t+1})\right\}.
\end{aligned} \tag{19}$$

Suppose $\mathcal{L}_{\mathrm{adv}}(\theta^{t+1}) > \mathcal{L}_{\mathrm{adv}}(\theta^t)$, and let $M = \left[(1+c)^2 + \frac{2c}{\eta^0\gamma_2}\right]$. We have

$$\mathcal{L}_{\mathrm{adv}}(\theta^{t+1}) \leq \mathcal{L}_{\mathrm{adv}}(\theta^t) - \eta^t||\nabla\mathcal{L}_{\mathrm{adv}}(\theta^t)||_2^2 + \frac{(\eta^t||\nabla\mathcal{L}_{\mathrm{adv}}(\theta^t)||)^2}{2}M\mathcal{L}_{\mathrm{adv}}(\theta^{t+1}),$$

which implies

$$\mathcal{L}_{\mathrm{adv}}(\theta^{t+1}) \leq \left(1 - \frac{M(\eta^t)^2}{2}||\nabla\mathcal{L}_{\mathrm{adv}}(\theta^t)||_2^2\right)^{-1}\left(\mathcal{L}_{\mathrm{adv}}(\theta^t) - \eta^t||\nabla\mathcal{L}_{\mathrm{adv}}(\theta^t)||_2^2\right). \tag{20}$$

Meanwhile, if we choose $\eta^t$ satisfying

$$\eta^t M = \eta^t \mathcal{L}_{\mathrm{adv}}(\theta^t)\left[(1+c)^2 + \frac{2c}{\eta^0\gamma_2}\right] \leq 1, \tag{21}$$

then we have the right hand side of (20) is upper bounded by $\mathcal{L}_{\mathrm{adv}}(\theta^t)$, and we have

$$\mathcal{L}_{\mathrm{adv}}(\theta^{t+1}) \leq \left(1 - \frac{M(\eta^t)^2}{2}||\nabla\mathcal{L}_{\mathrm{adv}}(\theta^t)||_2^2\right)^{-1}\left(\mathcal{L}_{\mathrm{adv}}(\theta^t) - \eta^t||\nabla\mathcal{L}_{\mathrm{adv}}(\theta^t)||_2^2\right) < \mathcal{L}_{\mathrm{adv}}(\theta^t),$$

which is clearly a contradiction. Hence, if $\eta^t$ satisfies (21), by (19) we have

$$\begin{aligned}
\mathcal{L}_{\mathrm{adv}}(\theta^{t+1}) &\leq \mathcal{L}_{\mathrm{adv}}(\theta^t) - \eta^t||\nabla\mathcal{L}_{\mathrm{adv}}(\theta^t)||_2^2 + \frac{(\eta^t||\nabla\mathcal{L}_{\mathrm{adv}}(\theta^t)||)^2}{2}\left[(1+c)^2 + \frac{2c}{\eta^0\gamma_2}\right]\mathcal{L}_{\mathrm{adv}}(\theta^t) \\
&\leq \mathcal{L}_{\mathrm{adv}}(\theta^t) - \frac{\eta^t}{2}||\nabla\mathcal{L}_{\mathrm{adv}}(\theta^t)||_2^2,
\end{aligned} \tag{22}$$

where the last inequality holds by the choice of $\eta^t$ in (21).

Note that if (21) holds for $t = 1$ for $\eta^1 = \eta$, by induction it is easy to see that with constant stepsize $\eta^t = \eta$ for $t \geq 1$, (21) holds for all $t \geq 1$. Hence for $t \geq 1$, we choose stepsize $\eta$ such that $\eta\mathcal{L}_{\mathrm{adv}}(\theta^1)\left[(1+c)^2 + \frac{2c}{\eta^0\gamma_2}\right] \leq 1$. Note that $\mathcal{L}_{\mathrm{adv}}(\theta^1) = \frac{1}{n}\sum_{i=1}^n \exp\left(-z_i^\top\theta^1 + c||\theta^1||_2\right) \leq \exp\left((1+c)\eta^0\right)$ since $||\theta_1|| \leq \eta^0$. Then we only require

$$\begin{aligned}
\eta &\leq \exp\left(-(1+c)\eta^0\right) \cdot \frac{\eta^0\gamma_2}{(1+c)^2\eta^0\gamma_2 + 2c} \\
&= \exp\left(-(1+c)\eta^0\right) \cdot \frac{\eta^0(1+c)\gamma_2/(1+c)}{(1+c)^2\eta^0\gamma_2 + 2c} \\
&\leq \frac{\gamma_2/e}{(1+c)^3\gamma_2 + 2c(1+c)},
\end{aligned}$$

where in the last inequality we take $\eta^0 = 1$ and use basic inequality $\exp(-x)x \leq e^{-1}$ for $x \geq 1$. In summary, we choose $\eta^0 = 1$ and $\eta^t = \eta = \min\{\frac{\gamma_2/e}{(1+c)^3\gamma_2 + 2c(1+c)}, 1\}$ for $t \geq 1$, then by previous argument, we have (22) holds for all $t \geq 1$.

Now we are ready to apply the standard smoothness-based analysis of gradient descent using (22), take any $\theta \in \mathbb{R}^d$, we have

$$\begin{aligned}
||\theta^{t+1} - \theta||_2^2 &= ||\theta^t - \theta||_2^2 - 2\eta^t\left\langle\nabla\mathcal{L}_{\mathrm{adv}}(\theta^t), \theta^t - \theta\right\rangle + (\eta^t)^2||\nabla\mathcal{L}_{\mathrm{adv}}(\theta^t)||_2^2 \\
&\leq ||\theta^t - \theta||_2^2 - 2\eta^t\left(\mathcal{L}_{\mathrm{adv}}(\theta^t) - \mathcal{L}_{\mathrm{adv}}(\theta)\right) + (\eta^t)^2||\nabla\mathcal{L}_{\mathrm{adv}}(\theta^t)||_2^2 \\
&\leq ||\theta^t - \theta||_2^2 - 2\eta^t\left(\mathcal{L}_{\mathrm{adv}}(\theta^t) - \mathcal{L}_{\mathrm{adv}}(\theta)\right) + 2\eta^t\left(\mathcal{L}_{\mathrm{adv}}(\theta^t) - \mathcal{L}_{\mathrm{adv}}(\theta^{t+1})\right) \\
&= ||\theta^t - \theta||_2^2 - 2\eta^t\left(\mathcal{L}_{\mathrm{adv}}(\theta^{t+1}) - \mathcal{L}_{\mathrm{adv}}(\theta)\right),
\end{aligned}$$

where the first inequality holds by the convexity of $\mathcal{L}_{\mathrm{adv}}(\theta)$, and the second inequality holds by (22). Now sum up the above inequality from $s = 1$ to $t - 1$. By $\eta^t = \eta \le 1 = \eta^0$ and $\mathcal{L}_{\mathrm{adv}}(\theta^{s+1}) \le \mathcal{L}_{\mathrm{adv}}(\theta^s)$, we have

$$\mathcal{L}_{\mathrm{adv}}(\theta^t) - \mathcal{L}_{\mathrm{adv}}(\theta) \le \frac{1}{2t\eta}||\theta^1 - \theta||_2^2 \le \frac{1}{t\eta}\left(||\theta||_2^2 + ||\theta^1||_2^2\right).$$

Now since $\theta$ is arbitrary, letting $\theta = \frac{\log(t)}{\gamma_2 - c} \cdot u_2$, we have

$$||\theta||_2^2 + ||\theta^1||_2^2 \le \frac{\log^2 t}{(\gamma_2 - c)^2} + (1 + c)^2,$$

and

$$\mathcal{L}_{\mathrm{adv}}(\theta) = \frac{1}{n}\sum_{i=1}^n \exp\left(-z_i^\top u_2 \cdot \frac{\log t}{\gamma_2 - c} + c \cdot \frac{\log t}{\gamma_2 - c}\right) \le \frac{1}{t},$$

which yields

$$\mathcal{L}_{\mathrm{adv}}(\theta^t) \le \frac{1}{t} + \left(\frac{\log^2 t}{(\gamma_2 - c)^2} + (1 + c)^2\right) = \mathcal{O}\left(\frac{\log^2 t}{t\eta(\gamma_2 - c)^2}\right).$$

$\square$

*Proof of Lemma 3.2.* For simplicity, we let $z_i = y_i x_i$ and $\ell_i(\theta) = \exp\left(-z_i^\top\theta + c||\theta||_2\right)$. Define

$$\alpha = \min_{||\xi||_2 = 1, \xi \in \mathrm{span}(u_2)^\perp} \max_{i \in \mathrm{SV}(\mathcal{S})} \langle\xi, z_i\rangle$$

where $\mathrm{SV}(\mathcal{S})$ denotes the set of support vectors. It has been shown in Ji and Telgarsky (2019) (Lemma 2.10) that $\alpha > 0$ with probability 1 if the data is sampled from absolutely continuous distribution.

We have

$$
\begin{aligned}
\langle\nabla\mathcal{L}_{\mathrm{adv}}(\theta^t), \theta_\perp^t\rangle &= \frac{1}{n}\left\langle\sum_{i=1}^n \exp\left(-z_i^\top\theta^t + c||\theta^t||_2\right)\left(-z_i + c\frac{\theta^t}{||\theta^t||_2}\right), \theta_\perp^t\right\rangle \\
&= \frac{1}{n}\sum_{i=1}^n \ell_i(\theta^t)\langle-z_i, \theta_\perp^t\rangle + \frac{1}{n}\sum_{i=1}^n \ell_i(\theta^t)\left\langle c\frac{\theta^t}{||\theta^t||_2}, \theta_\perp^t\right\rangle \\
&\ge \frac{1}{n}\sum_{i=1}^n \ell_i(\theta^t)\langle-z_i, \theta_\perp^t\rangle \\
&\ge \frac{1}{n}\left[\ell_j(\theta^t)\langle-z_j', \theta_\perp^t\rangle + \sum_{\langle z_i, \theta_\perp^t\rangle \ge 0, i \ne j} \ell_i(\theta^t)\langle-z_i, \theta_\perp^t\rangle\right],
\end{aligned}
$$

(23)

where $z_j' \in S$ is arbitrary, by definition of $\alpha$: $\langle-z_j', \theta_\perp^t\rangle \ge \alpha||\theta_\perp^t||_2$.

We bound the first term as

$$
\begin{aligned}
\ell_j(\theta^t)\langle-z_j', \theta_\perp^t\rangle &\ge \exp\left(-(z_j')^\top\theta^t + c||\theta^t||_2\right)\alpha||\theta_\perp^t||_2 \\
&= \exp\left(-(z_j')^\top\theta_\perp^t - (z_j')^\top\theta_{u_2}^t + c||\theta^t||_2\right)\alpha||\theta_\perp^t||_2 \\
&\ge \exp\left(-\langle\theta^t, \gamma_2 u_2\rangle\right)\exp\left(\alpha||\theta_\perp^t||_2\right)\alpha||\theta_\perp^t||_2\exp\left(c||\theta^t||_2\right),
\end{aligned}
$$

where the second inequality uses $\langle z_j', u_2\rangle \ge \gamma_2$.

On the other hand, we can bound the second term in (23) as

$$
\begin{aligned}
\frac{1}{n}\sum_{\langle z_i, \theta_\perp^t\rangle \ge 0, i \ne j} \ell_i(\theta^t)\langle-z_i, \theta_\perp^t\rangle &\ge \frac{1}{n}\sum_{\langle z_i, \theta_\perp^t\rangle \ge 0, i \ne j}\exp\left(-z_i^\top\theta^t + c||\theta^t||_2\right)\langle-z_i, \theta_\perp^t\rangle \\
&= \frac{1}{n}\sum_{\langle z_i, \theta_\perp^t\rangle \ge 0, i \ne j}\exp\left(-z_i^\top\theta_{u_2}^t - z_i^\top\theta_\perp^t + c||\theta^t||_2\right)\langle-z_i, \theta_\perp^t\rangle \\
&\ge \exp\left(-\langle\theta^t, \gamma_2 u_2\rangle\right)\exp(c||\theta^t||_2)\exp(-z_i^\top\theta_\perp^t)\langle-z_i, \theta_\perp^t\rangle \\
&\ge \exp\left(-\langle\theta^t, \gamma_2 u_2\rangle\right)\exp(c||\theta^t||_2)(-\frac{1}{e}),
\end{aligned}
$$

where in the last inequality holds since $\langle \theta^t, u_2 \rangle \geq 0$, $\langle z_i, \theta_{u_2}^t \rangle = z_i^\top \left( u_2^\top \theta^t \right) u_2 \geq \gamma_2 \langle \theta^t, u_2 \rangle$ and $-x \exp(-x) \geq -\frac{1}{e}$ for $x \geq 0$.

Plugging the two bounds above into (23), we have

$$\langle \nabla \mathcal{L}_{\mathrm{adv}}(\theta^t), \theta_\perp^t \rangle \geq \exp\left( -\langle \theta^t, \gamma_2 u_2 \rangle \right) \exp(c||\theta^t||_2) \left[ \frac{1}{n} \exp\left( \alpha ||\theta_\perp^t||_2 \right) \alpha ||\theta_\perp^t||_2 - \frac{1}{e} \right],$$

which is non-negative when $||\theta_\perp^t||_2 \geq K' = \frac{1 + \log n}{\alpha}$.

Supposing $||\theta_\perp^t||_2 \geq K'$, by gradient descent update, we have,
$$||\theta_\perp^{t+1}||_2^2 = ||\theta_\perp^t||_2^2 - 2\eta^t \langle \nabla \mathcal{L}_{\mathrm{adv}}(\theta^t), \theta_\perp^t \rangle + (\eta^t)^2 ||\nabla \mathcal{L}_{\mathrm{adv}}(\theta^t)||^2$$
$$\leq ||\theta_\perp^t||_2^2 + 2\eta^t ||\nabla \mathcal{L}_{\mathrm{adv}}(\theta^t)||_2^2$$
$$\leq ||\theta_\perp^t||_2^2 + 2 \left( \mathcal{L}_{\mathrm{adv}}(\theta^t) - \mathcal{L}_{\mathrm{adv}}(\theta^{t+1}) \right), \tag{24}$$
where the last inequality uses (22).

Now let $t_0$ satisfy $||\theta_\perp^{t_0-1}||_2 < K'$ and $||\theta_\perp^{t_0-1}||_2 \geq K'$. Define $t_1 = \min\{s \geq t_0 : ||\theta_\perp^s||_2 < K'\}$, when $||\theta_\perp^s||_2 \geq K'$ for all $s \geq t_0$ we define $t_1 = \infty$. That is for any $t \in \{t_0, \ldots, t_1 - 1\}$, we have $||\theta_\perp^t||_2 \geq K'$. then for any $s$ such that $t_0 \leq s < t_1$, summing (24) up from $t_0$ to $s - 1$ yields:
$$||\theta_\perp^s||_2^2 \leq ||\theta_\perp^{t_0}||_2^2 + 2 \left( \mathcal{L}_{\mathrm{adv}}(\theta^{t_0}) - \mathcal{L}_{\mathrm{adv}}(\theta^s) \right)$$
$$\leq ||\theta_\perp^t||_2^2 + 2 \exp(1 + c)$$
$$\leq ||\theta_\perp^t||_2^2 + 18,$$
where we use $\mathcal{L}_{\mathrm{adv}}(\theta^t) \leq \mathcal{L}_{\mathrm{adv}}(\theta^1) \leq \exp(1 + c)$ and $c < 1$. This inequality shows that for $\theta^t \in \{\theta^{t_0}, \ldots, \theta^{t_1-1}\} \subset \{\theta : ||\theta_\perp||_2 \geq K'\}$,
$$||\theta_\perp^t||_2 \leq ||\theta_\perp^{t_0}||_2 + 18.$$
Then, we only need to bound $||\theta_\perp^{t_0}||_2$ to conclude the proof, where $t_0$ is the first time $\theta^t$ enters $\{\theta : ||\theta_\perp||_2 \geq K'\}$. We have

$$\theta_\perp^{t_0} = \theta_\perp^{t_0-1} + \eta^{t_0-1} P_\perp \left( \frac{1}{n} \sum_{i=1}^n \ell_i(\theta^{t_0-1})(z_i - c\frac{\theta^{t_0-1}}{||\theta^{t_0-1}||_2}) \right),$$

where $P_\perp(\cdot)$ denotes the projection onto $\mathrm{span}(u_2)^\perp$. Note that $t_0$ is the first time $\theta^t$ (re)-enters the region $\{\theta : ||\theta_\perp||_2 \geq K'\}$, and thus $||\theta_\perp^{t_0-1}||_2 < K'$. We have
$$||\theta_\perp^{t_0}||_2 \leq K' + \eta^{t_0-1}(1 + c) \leq K' + 1 + c < K' + 2,$$
where the last inequality we use $c < \gamma_2 \leq 1$.

In summary, we have shown that for any $t$ such that $||\theta_\perp^t||_2 \geq K'$, we have $||\theta_\perp^t||_2 \leq K' + 20$, and we conclude that $||\theta_\perp^t||_2 = K' + 20 = K$ for all $t \geq 0$. Note that $K$ only depends $\alpha(\mathcal{S})$ and sample size $n$. $\qquad \square$

*Proof of Lemma 3.3.* To obtain a lower bound on $||\theta^t||_2$, we first denote $\theta^t = \theta_u^t + \theta_\perp^t$, where $\theta_u^t$ denotes the projection of $\theta$ onto $\mathrm{span}(u_2)$, and $\theta_\perp^t$ denotes the projection of $\theta$ onto $\mathrm{span}(u_2)^\perp$. We have

$$\frac{1}{n} \sum_{i=1}^n \exp(-z_i^\top \theta_u^t - z_i^\top \theta_\perp^t) \leq \frac{\log^2 t}{t \eta (\gamma_2 - c)^2} \exp(-c||\theta^t||_2).$$

Let us assume that $||\theta_\perp^t||$ is bounded so that $\exp(||\theta_\perp^t||) \leq M$, which will be verified immediately. Choosing an arbitrary support vector $z_i$, we have $0 < \langle z_i, \theta_u^t \rangle = \langle z_i, u_2 \rangle \langle \theta^t, u_2 \rangle = \gamma_2 \langle \theta^t, u_2 \rangle = \gamma_2 ||\theta_u^t||_2 \leq \gamma_2 ||\theta^t||_2$, hence the previous inequality becomes:

$$\exp(-\gamma_2 ||\theta^t||_2) \leq \frac{n \log^2 t}{t \eta (\gamma_2 - c)^2} \exp(-c||\theta^t||_2) M,$$

which is equivalent to

$$||\theta^t||_2 \geq \log \left( \frac{t \eta (\gamma_2 - c)^2}{n M \log^2 t} \right) / (\gamma_2 - c). \tag{25}$$

Now we only need to show that $||\theta_\perp^t|| \leq M$ for all $t$ for some $M$. Since we have shown in Lemma 3.2 that $||\theta^t||_2 \leq K$, we choose $M = e^K \leq \exp\left( \frac{20 + \log n}{\alpha} \right) = \mathcal{O}(n^{\frac{1}{\alpha}})$, and the lower bound (25) becomes

$$||\theta^t||_2 \geq \log \left( \frac{t \eta (\gamma_2 - c)^2}{n^{1+1/\alpha} \log^2 t} \right) / (\gamma_2 - c), \tag{26}$$

which concludes our proof. $\qquad \square$

*Proof of Theorem 3.2.* We denote $\theta^t = \theta_u^t + \theta_\perp^t$, where $\theta_u^t$ denotes the projection of $\theta$ onto $\mathrm{span}(u_2)$, and $\theta_\perp^t$ denotes the projection of $\theta$ onto $\mathrm{span}(u_2)^\perp$. Combine Lemma 3.2 and Lemma 3.3, we have

$$
\begin{aligned}
1 - \left\langle \frac{\theta^t}{||\theta^t||_2}, u_2 \right\rangle = 1 - \frac{\langle \theta_{u_2}^t, u_2 \rangle + \langle \theta_\perp^t, u_2 \rangle}{||\theta^t||_2} &\leq 1 - \frac{\langle \theta_{u_2}^t, u_2 \rangle}{||\theta^t||_2} + \frac{K}{||\theta^t||_2} \\
&= 1 - \frac{||\theta_{u_2}^t||_2}{||\theta^t||_2} + \frac{K}{||\theta^t||_2} \leq 1 - \frac{||\theta_{u_2}^t||_2^2}{||\theta^t||_2^2} + \frac{K}{||\theta^t||_2} \\
&= \frac{||\theta_\perp^t||_2^2}{||\theta^t||_2^2} + \frac{K}{||\theta^t||_2} \\
&\leq \frac{K^2}{||\theta^t||_2^2} + \frac{K}{||\theta^t||_2}.
\end{aligned}
$$

By our choice of $c$ and $T$ that $\gamma_2 - c = \left( \frac{n^{1+1/\alpha} \log^2 T}{\eta T} \right)^{1/2}$, together Lemma 3.3, the Theorem holds as desired. $\qquad\square$

*Proof of Corollary 3.2.* By Lemma 3.3 and the the choice of parameters that $\gamma_2 - c = \left( \frac{n^{1+1/\alpha} \log^2 T}{\eta T} \right)^{1/2}$, we have:

$$
||\theta^T||_2 \geq \left( \frac{\eta T}{n^{(1+1/\alpha)} \log^2 T} \right)^{1/2}.
$$

Together with Theorem 3.1, we have

$$
\begin{aligned}
\mathcal{L}(\theta^T) = \mathcal{L}_{\mathrm{adv}}(\theta^T) \exp\left( -c ||\theta^T||_2 \right) \\
\leqslant \frac{\log^2 T}{T \eta (\gamma_2 - c)^2} \exp\left( -c \left( \frac{\eta T}{n^{(1+1/\alpha)} \log^2 T} \right)^{1/2} \right) \\
= \mathcal{O}\left( \exp\left( -c \left( \frac{\eta T}{n^{(1+1/\alpha)} \log^2 T} \right)^{1/2} \right) \right).
\end{aligned}
$$

where the last equality holds by the parameter choice $\gamma_2 - c = \left( \frac{n^{1+1/\alpha} \log^2 T}{\eta T} \right)^{1/2}$. Finally, letting $\mu = c \left( \frac{\eta}{n^{1+1/\alpha}} \right)^{1/2}$, the claim follows immediately. $\qquad\square$

## C  PROOFS FOR SECTION 3.2

In this section, we consider general $\ell_q$-norm perturbations. In short, we show that no matter how small the perturbation is, adversarial training changes the implicit bias of standard clean training using gradient descent, and adapt it to specific norm we choose for adversarial training.

Intuitively, we might expect that under the $\ell_q$-norm perturbation the implicit bias of gradient descent algorithm changes to converging in direction to $\ell_q$-norm max margin solution $\bar{u}_q$. We provide a counter example here. Consider $\mathcal{S} = \{z_1 = (x_1, y_1), z_2 = (x_2, y_2)\}$ with $x_1 = (10, 1), x_2 = (-10, -1)$ and $y_1 = 1, y_2 = -1$.

It is easy to see that the $\ell_\infty$-norm max margin solution is $\bar{u}_\infty = (1, 0)$ with $\gamma_\infty = 10$, and the $\ell_2$-norm max margin solution is $\bar{u}_2 = (\frac{10}{\sqrt{101}}, \frac{1}{\sqrt{101}})$ with $\gamma_2 = \sqrt{101}$.

Without perturbation, we have that the gradient descent initialized at the origin converges in direction to $\ell_2$-norm max margin solution $\bar{u}_2$ with one step. Now we take $l_\infty$-norm perturbation with $c = 0.5$, the negative gradient is given by: $-\nabla \mathcal{L}_{\mathrm{adv}}(\theta) = \frac{\ell_1(\theta)}{2}(z_1 - c \cdot \mathrm{sign}(\theta)) + \frac{\ell_2(\theta)}{2}(z_2 - c \cdot \mathrm{sign}(\theta))$. We initialize gradient descent at the origin with any constant step size. By the symmetry of the training data, we have that $\theta^t$ always stays always inside quadrant I, and converges in direction to $\bar{u} = (\frac{\sqrt{361}}{\sqrt{362}}, \frac{1}{\sqrt{362}})$, which is neither $\bar{u}_\infty$ or $\bar{u}_2$, but inside the interior of convex hull of $\bar{u}_\infty$ and $\bar{u}_2$. In fact, $\bar{u}$ exactly equals to the $u_{2,\infty}$ defined in (10).

*Proof of Lemma 3.4.* We prove that solutions to (10) and the robust SVM against $\ell_q$-norm perturbation parameterized by $c$ (8) are equal up to a constant factor. We first have that $\gamma_{2,q}(c)$ in (10) is equivalent to

$$\gamma_{2,q} = \max_{||\theta||_2 \leq 1} \min_{i \in [n]} y_i x_i^\top \theta - c||\theta||_p. \tag{27}$$

We denote the unique solution to (27) as $u_{2,q}$. It is not difficulty to see that

$$y_i x_i^\top u_{2,q} - c||u_{2,q}||_2 \geqslant \gamma_{2,q}, \forall i = 1, \ldots, n.$$

We define $\overline{u}_{2,q} = \frac{u_{2,q}}{\gamma_{2,q}}$, then:

$$y_i x_i^\top \overline{u}_{2,q} - c||\overline{u}_{2,q}||_2 \geqslant 1, \forall i = 1, \ldots, n.$$

It is now clear that $\overline{u}_{2,q}$ is a feasible solution to (8). We denote the optimal solution to (8) as $\overline{u}$, then we have by the optimality of $\overline{u}$ that $||\overline{u}||_2 \leq ||\overline{u}_{2,q}||_2 \leq \frac{||u_{2,q}||_2}{\gamma_{2,q}}$, and feasibility of $\overline{u}$ that

$$y_i x_i^\top (\gamma_{2,q} \overline{u}) - c||\gamma_{2,q} \overline{u}||_2 \geq \gamma_{2,q} \forall i = 1, \ldots, n.$$

Then from previous two inequalities we have $\gamma_{2,q} \overline{u}$ is a feasible solution to (27) with objective value equal to the optimal objective value of (27). Since the optimal solution to (27) is unique, this implies that $\overline{u} = \frac{u_{2,q}}{\gamma_{2,q}}$, which concludes our proof. $\qquad\square$

We extend Lemma 3.1 to bounded $\ell_q$-norm perturbation set.

**Lemma C.1.** *Recall the definition of $\gamma_{2,q}$ in (10). For any $c < \gamma_q$, we have that $\langle -\nabla \mathcal{L}_{adv}(\theta), u_{2,q} \rangle \geq \mathcal{L}_{adv}(\theta)\gamma_{2,q}$ for all $\theta \in \mathbb{R}^d$.*

*Proof.* Recall that we have $\mathcal{L}_{adv}(\theta) = \frac{1}{n}\sum_{i=1}^n \max_{||\delta||_q \leq c} \exp\left(-y_i(x_i + \delta_i)^\top \theta\right)$. For each sample $(x_i, y_i) \in \mathcal{S}$, given a classifier $\theta$, the worst case perturbation is $\widetilde{\delta}_i = \text{argmax}_{||\delta||_q \leq c} \exp\left(-y_i(x_i + \delta)^\top \theta\right) = \text{argmin}_{||\delta||_q \leq c} y_i \delta^\top \theta$. The corresponding loss is then $\mathcal{L}_{adv}(\theta) = \frac{1}{n}\sum_{i=1}^n \exp\left(-y_i(x_i + \widetilde{\delta}_i)^\top \theta\right)$.

Since for a fixed $\delta_i$, the function $\exp\left(-y_i(x_i + \delta_i)^\top \theta\right)$ is convex in $\theta$, hence the gradient of $\mathcal{L}_{adv}(\theta)$ is

$$-\nabla \mathcal{L}_{adv}(\theta) = \frac{1}{n}\sum_{i=1}^n \exp\left(-y_i(x_i + \widetilde{\delta}_i)^\top \theta\right) y_i(x_i + \widetilde{\delta}_i).$$

Then by the definition of $u_{2,q}$, we have

$$\langle -\nabla \mathcal{L}_{adv}(\theta), u_2 \rangle = \sum_{i=1}^n \exp\left(-y_i(x_i + \widetilde{\delta}_i)^\top \theta\right) \left\langle y_i(x_i + \widetilde{\delta}_i), u_{2,q} \right\rangle \tag{28}$$

$$\geq \sum_{i=1}^n \exp\left(-y_i(x_i + \widetilde{\delta}_i)^\top \theta\right) \gamma_{2,q} = \mathcal{L}_{adv}(\theta)\gamma_{2,q}, \tag{29}$$

where the second inequality holds by $||\widetilde{\delta}_i||_q \leq c$, and the definitions of $u_{2,q}$ and $\gamma_{2,q}$ in Lemma 3.4. $\qquad\square$

Note that for $q = 2$, by the fact that $\gamma_{2,2}(c) = \gamma_2 - c$, we immediately have Lemma 3.1 holds.

As a direct corollary of Lemma C.1, we have $||\theta^t||_2$ is bounded away from 0 for all $t \geq 1$.

**Corollary C.1.** *Let $\theta^0 = 0$ in Algorithm 1, we have: $||\theta^t||_2 \geq \eta^0 \gamma_{2,q}$ for al $t \geq 1$.*

*Proof.* The proof is similar to Corollary 3.1, we omit the details here. $\qquad\square$

*Proof of Theorem 3.3.* For simplicity, we define $z_i = y_i x_i$ and have $||z_i||_2 \leq 1$ since $||x_i||_2 \leq 1$. We have for $\theta \neq 0$

$$\nabla \mathcal{L}_{adv}(\theta) = \frac{1}{n}\sum_{i=1}^n \exp\left(-z_i^\top \theta + c||\theta||_p\right)(-z_i + c\partial||\theta||_p),$$

$$\nabla^2 \mathcal{L}_{adv}(\theta) = \frac{1}{n}\sum_{i=1}^n \exp\left(-z_i^\top \theta + c||\theta||_p\right)(-z_i + c\partial||\theta||_p)(-z_i + c\partial||\theta||_p)^\top$$

$$+ \frac{1}{n}\sum_{i=1}^n \exp\left(-z_i^\top \theta + c||\theta||_p\right) c\left((1-p)||\theta||_p^{1-2p}(\odot^{p-1}\theta)(\odot^{p-1}\theta)^\top + (p-1)||\theta||_p^{1-p}\text{diag}(\odot^{p-2}\theta)\right),$$

where $\odot^{p-1}\theta$ denotes taking element-wise $(p-1)$-th power of $\theta$.

Note that we have $||\partial||\theta||_p||_q = 1$. By the conjugacy of $\ell_p$-norm and $\ell_q$-norm with $\frac{1}{p} + \frac{1}{q} = 1$, we have $||\theta||_p = \max_{||s||_q \leq 1} \langle \theta, s \rangle$. Hence we upper bound the first term in Hessian $\nabla^2 \mathcal{L}_{\mathrm{adv}}(\theta)$ above by

$$\frac{1}{n} \sum_{i=1}^{n} \exp\left(-z_i^\top \theta + c||\theta||_p\right) \left(-z_i + c\partial||\theta||_p\right) \left(-z_i + c\partial||\theta||_p\right)^\top \tag{30}$$

$$\leq \frac{1}{n} \sum_{i=1}^{n} \exp\left(-z_i^\top \theta + c||\theta||_p\right) (1 + c\sqrt{d}||\theta||_2)^2. \tag{31}$$

We further have:

$$(p-1)||\theta||_p^{p-1}\mathrm{diag}(\odot^{p-2}\theta) \leq (p-1)\frac{\mathrm{diag}(\odot^{p-2}\theta)}{d^{\frac{p}{p-1}}||\theta||_\infty^{p-1}}$$

$$\leq (p-1)d^{\frac{p}{p-1}}\frac{I}{||\theta||_\infty}$$

$$\leq (p-1)d^{\frac{3p-2}{2p-2}}\frac{I}{||\theta||_2}.$$

Together with the fact that $p \geq 1$, we bound the Hessian $\nabla^2 \mathcal{L}_{\mathrm{adv}}(\theta)$ as:

$$\nabla^2 \mathcal{L}_{\mathrm{adv}}(\theta) \leq \mathcal{L}_{\mathrm{adv}}(\theta)\left[(1 + c\sqrt{d})^2 + c(p-1)d^{\frac{3p-2}{2p-2}}\frac{1}{||\theta||_2}\right] I.$$

Note that the Hessian expression indicates that the objective is highly non-smooth around origin. However, as shown in Corollary C.1, starting from origin, $\theta^t$ always stays away from the origin with distance bounded below.

Using Taylor expansion, and by $\theta^{t+1} = \theta^t - \eta^t \nabla \mathcal{L}_{\mathrm{adv}}(\theta^t)$, we have

$$\mathcal{L}_{\mathrm{adv}}(\theta^{t+1}) \leq \mathcal{L}_{\mathrm{adv}}(\theta^t) - \eta^t||\nabla \mathcal{L}_{\mathrm{adv}}(\theta^t)||_2^2 + \frac{(\eta^t||\nabla \mathcal{L}_{\mathrm{adv}}(\theta^t)||)^2}{2} \max_{v \in [\theta^t, \theta^{t+1}]} \lambda\left(H(v)\right)_{\mathrm{max}}, \tag{32}$$

where $\lambda\left(H(v)\right)_{\mathrm{max}}$ denotes the largest eigenvalue of $H(v)$, and

$$H(v) = \mathcal{L}_{\mathrm{adv}}(v)\left[(1 + c\sqrt{d})^2 + c(p-1)d^{\frac{3p-2}{2p-2}}\frac{1}{||v||_2}\right] I.$$

Since $\eta^0 = 1$, by Corollary C.1, for any $t \geq 1$, we have $||\theta^t||_2 \geqslant \gamma_{2,q}$. Letting $m_p = (1 + c\sqrt{d})^2 + c(p-1)d^{\frac{3p-2}{2p-2}}\frac{1}{\gamma_{2,q}}$, and since that $\mathcal{L}_{\mathrm{adv}}(\theta)$ is a convex function, we obtain that

$$\mathcal{L}_{\mathrm{adv}}(\theta^{t+1}) \leq \mathcal{L}_{\mathrm{adv}}(\theta^t) - \eta^t||\nabla \mathcal{L}_{\mathrm{adv}}(\theta^t)||_2^2 + \frac{(\eta^t||\nabla \mathcal{L}_{\mathrm{adv}}(\theta^t)||)^2}{2}m_p \max\{\mathcal{L}_{\mathrm{adv}}(\theta^{t+1}), \mathcal{L}_{\mathrm{adv}}(\theta^t)\}.$$

We then show by contradiction that we have $\mathcal{L}_{\mathrm{adv}}(\theta^{t+1}) < \mathcal{L}_{\mathrm{adv}}(\theta^t)$. Assume this is not the case, then we have:

$$\mathcal{L}_{\mathrm{adv}}(\theta^{t+1}) \leq \left(1 - \frac{M(\eta^t)^2}{2}||\nabla \mathcal{L}_{\mathrm{adv}}(\theta^t)||_2^2\right)^{-1}\left(\mathcal{L}_{\mathrm{adv}}(\theta^t) - \eta^t||\nabla \mathcal{L}_{\mathrm{adv}}(\theta^t)||_2^2\right)$$

However, if we choose $\eta^t$ satisfying $\eta^t \leq \frac{2}{m_q \mathcal{L}_{\mathrm{adv}}(\theta^t)}$, we have the right hand side of previous inequality strictly smaller than $\mathcal{L}_{\mathrm{adv}}(\theta^t)$, which is clearly a constradiction. Hence when we choose $\eta^t \leq \frac{2}{m_q \mathcal{L}_{\mathrm{adv}}(\theta^t)}$, we have $\mathcal{L}_{\mathrm{adv}}(\theta^{t+1}) < \mathcal{L}_{\mathrm{adv}}(\theta^t)$ and

$$\mathcal{L}_{\mathrm{adv}}(\theta^{t+1}) \leq \mathcal{L}_{\mathrm{adv}}(\theta^t) - \eta^t||\nabla \mathcal{L}_{\mathrm{adv}}(\theta^t)||_2^2 + \frac{(\eta^t||\nabla \mathcal{L}_{\mathrm{adv}}(\theta^t)||)^2}{2}m_p \mathcal{L}_{\mathrm{adv}}(\theta^t). \tag{33}$$

Now by induction, if we choose $\eta^t = \eta \leq \frac{1}{m_q \mathcal{L}_{\mathrm{adv}}(\theta^1)}$ for $t \geq 1$, then we have (33) holds for all $t \geq 1$. Note that we have an upper bound of $\mathcal{L}_{\mathrm{adv}}(\theta^1)$, which is

$$\mathcal{L}_{\mathrm{adv}}(\theta^1) = \frac{1}{n} \sum_{i=1}^{n} \exp\left(-y_i(x_i + \widetilde{\delta}_i)^\top \theta^1\right)$$

$$= \frac{1}{n} \sum_{i=1}^{n} \exp\left(-y_i(x_i + \widetilde{\delta}_i)^\top \theta_u^1 - y_i(x_i + \widetilde{\delta}_i)^\top \theta_\perp^1\right)$$

$$\leq \frac{1}{n} \sum_{i=1}^{n} \exp\left(-\gamma_{2,q}^2 + (1 + c\sqrt{d})\right) = \exp\left(-\gamma_{2,q}^2 + (1 + c\sqrt{d})\right), \tag{34}$$

where $\widetilde{\delta}_i$ denotes the worst case perturbation to $x_i$, and $\theta_u^1$ denotes projection of $\theta^1$ onto $\text{span}(u_{2,q})$, and $\theta_\perp$ denotes projection of $\theta^1$ onto $\text{span}(u_{2,q})^\perp$.

In summary, we have that if

$$\eta^t = \eta \leq \min\{\frac{1}{M_p}, 1\} \text{ for all } t \geq 1, \text{ where } M_p = m_p \exp\left(-\gamma_{2,q}^2 + (1 + c\sqrt{d})\right), \tag{35}$$

we have

$$\mathcal{L}_{\text{adv}}(\theta^{t+1}) \leq \mathcal{L}_{\text{adv}}(\theta^t) - \eta||\nabla\mathcal{L}_{\text{adv}}(\theta^t)||_2^2 + \frac{(\eta||\nabla\mathcal{L}_{\text{adv}}(\theta^t)||)^2}{2}m_p\mathcal{L}_{\text{adv}}(\theta^t) \tag{36}$$

$$\leq \mathcal{L}_{\text{adv}}(\theta^t) - \frac{\eta}{2}||\nabla\mathcal{L}_{\text{adv}}(\theta^t)||_2^2 \tag{37}$$

where the last inequality holds since $\eta m_p \mathcal{L}_{\text{adv}}(\theta^t) \leq \eta m_p \mathcal{L}_{\text{adv}}(\theta^1) \leq 1$. Now for any $\theta \in \mathbb{R}^d$, we have

$$||\theta^{t+1} - \theta||_2^2 = ||\theta^t - \theta||_2^2 - 2\eta^t\left\langle\nabla\mathcal{L}_{\text{adv}}(\theta^t), \theta^t - \theta\right\rangle + (\eta^t)^2||\nabla\mathcal{L}_{\text{adv}}(\theta^t)||_2^2$$

$$\leq ||\theta^t - \theta||_2^2 - 2\eta^t\left(\mathcal{L}_{\text{adv}}(\theta^t) - \mathcal{L}_{\text{adv}}(\theta)\right) + (\eta^t)^2||\nabla\mathcal{L}_{\text{adv}}(\theta^t)||_2^2$$

$$\leq ||\theta^t - \theta||_2^2 - 2\eta^t\left(\mathcal{L}_{\text{adv}}(\theta^t) - \mathcal{L}_{\text{adv}}(\theta)\right) + 2\eta^t\left(\mathcal{L}_{\text{adv}}(\theta^t) - \mathcal{L}_{\text{adv}}(\theta^{t+1})\right)$$

$$= ||\theta^t - \theta||_2^2 - 2\eta^t\left(\mathcal{L}_{\text{adv}}(\theta^{t+1}) - \mathcal{L}_{\text{adv}}(\theta)\right),$$

where the first inequality holds by the convexity of $\mathcal{L}_{\text{adv}}(\theta)$, and the second inequality holds by (37).

Summing up the above inequality from $s = 1$ to $t - 1$ and by $\eta^t = \eta \leq 1 = \eta^0$ together with $\mathcal{L}_{\text{adv}}(\theta^{s+1}) \leq \mathcal{L}_{\text{adv}}(\theta^s)$, we have

$$\mathcal{L}_{\text{adv}}(\theta^t) - \mathcal{L}_{\text{adv}}(\theta) \leq \frac{1}{2t\eta}||\theta^1 - \theta||_2^2 \leq \frac{1}{t\eta}\left(||\theta||_2^2 + ||\theta^1||_2^2\right) \tag{38}$$

Since $\theta$ is arbitrary, by choosing $\theta = \frac{\log(t)}{\gamma_{2,q}} \cdot u_{2,q}$, we have

$$||\theta||_2^2 + ||\theta^1||_2^2 \leq \frac{\log^2 t}{\gamma_{2,q}^2} + (1 + c\sqrt{d})^2,$$

and

$$\mathcal{L}_{\text{adv}}(\theta) = \frac{1}{n}\sum_{i=1}^{n}\exp\left(-\min_{||\delta_i||_q \leq c}(z_i + \delta_i)^\top u_{2,q}\frac{\log t}{\gamma_{2,q}}\right) \leq \frac{1}{t},$$

which yields

$$\mathcal{L}_{\text{adv}}(\theta^t) \leq \frac{1}{t} + \frac{1}{t\eta}\left(\frac{\log^2 t}{\gamma_{2,q}^2} + (1 + c\sqrt{d})^2\right) = \mathcal{O}\left(\frac{\log^2 t}{t\eta\gamma_{2,q}^2}\right). \tag{39}$$

$\square$

**Parameter Convergence: Intuition**. Before we formally prove the implicit bias of GDAT, we provide some intuitions here for better understanding. We claim that $\overline{u}_\infty = \lim_{t\to\infty}\frac{\theta^t}{||\theta||_2}$ is in the same direction as the solution to

$$\min_\theta \frac{1}{2}||\theta||_2 + \eta(c)||\theta||_p, \quad \text{s.t.} \quad z_i^\top\theta \geq 1, \forall i = 1, \ldots n. \tag{40}$$

Note that $\theta^t$ is a conic combination of $\{z_i - ca||\theta^t||_p\}_{i\in[n]}$, and $\partial||\theta^t||_p$ only depends on the direction of $\theta^t$. Hence by normalizing the norm of $\theta^t$ and using $\lim_{t\to\infty}||\theta^t||_2 = \infty$, if the limit $\overline{u}_\infty = \lim_{t\to\infty}\frac{\theta^t}{||\theta^t||_2}$ exists, it satisfies the following condition under proper scaling that

$$\theta = \sum_{i=1}^{n}a_i(z_i - c\partial||\theta^t||_p),$$

$$\text{s.t.} \quad a_i \geq 0, z_i^\top\theta \geq 1, \forall i = 1, \ldots n,$$

$$a_i(z_i^\top\theta - 1) = 0, \forall i = 1, \ldots n.$$

Defining $a = (a_1, \ldots, a_n)$ and $(\widehat{\theta}, a) = \left((||\theta||_p c + 1)\theta, (||\theta||_p c + 1)a\right)$, it is easy to see that $(\widehat{\theta}, a)$ is a solution to the following system

$$\theta = \sum_{i=1}^{n}a_i(z_i - c\partial||\theta^t||_p), \tag{41}$$

$$\text{s.t.:} \quad a_i \geq 0, z_i^\top\theta \geq c||\theta||_p + 1, \forall i = 1, \ldots n. \tag{42}$$

$$a_i(z_i^\top\theta - c||\theta||_p - 1) = 0, \forall i = 1, \ldots n. \tag{43}$$

Notice that the above set of equations (41)-(43) is exactly the first-order KKT condition of the following optimization problem

$$\min_{\theta} \frac{1}{2}||\theta||_2^2 \quad \text{s.t.} \quad z_i^\top \theta \geq c||\theta||_p + 1, \forall i = 1, \ldots n. \tag{44}$$

(44) has a robust reformulation as maximizing the $\ell_2$-norm margin under the worse case $\ell_q$-norm perturbation bounded by $c$ that

$$\min_{\theta} \frac{1}{2}||\theta||_2^2 \quad \text{s.t.} \quad \min_{||\delta_i||_q \leq c} (z_i + \delta_i)^\top \theta \geq 1, \forall i = 1, \ldots n,$$

or equivalently

$$\max_{\theta} \min_{i=1,\ldots,n} \min_{||\delta_i||_\infty \leq c} \frac{y_i(x_i + \delta_i)^\top \theta}{||\theta||_q}. \tag{45}$$

We note that (45) is a Support Vector Machine problem over an uncoutable data set that is generated by norm-bounded perturbation $\mathcal{S}(c,q) = \{(x,y): \text{where } \exists i \in [n], ||x - x_i||_q \leq c, y = y_i\}$. By the separability and $c < \gamma_q$, we have that $\mathcal{S}(c,q)$ is well defined.

By the first-order KKT condition we have that (44) is equivalent to

$$\min_{\theta} ||\theta||_2 + \eta(c)||\theta||_p \quad \text{s.t.} \quad z_i^\top \theta \geq 1, \forall i = 1, \ldots n.$$

for some proper $\eta(c)$ that depends on $c$. Hence in summary, if $\overline{u}_\infty = \lim_{t\to\infty} \frac{\theta^t}{||\theta^t||_2}$ exists, it is in the same direction as the solution to the mixed $(\ell_2, \ell_1)$-norm max margin solution of (40).

**Claim:** In general, for $\ell_q$-norm perturbation bounded by $c$, $\theta^t$ converges in direction to the solution to

$$\min_{\theta} \frac{1}{2}||\theta||_2^2 \quad \text{s.t.} \quad \min_{||\delta_i||_q \leq c} (z_i + \delta_i)^\top \theta \geq 1, \forall i = 1, \ldots n.$$

or

$$\min_{\theta} ||\theta||_2 + \eta(c)||\theta||_p \quad \text{s.t.} \quad z_i^\top \theta \geq 1, \forall i = 1, \ldots n.$$

for some proper $\eta(c)$ that depends on $c$.

*Proof of Theorem 3.4.* Recall that in Theorem 3.3 we showed in (36) the following recursion

$$\begin{aligned}
\mathcal{L}_{\text{adv}}(\theta^{t+1}) &\leq \mathcal{L}_{\text{adv}}(\theta^t) - \eta||\nabla\mathcal{L}_{\text{adv}}(\theta^t)||_2^2 + \frac{(\eta||\nabla\mathcal{L}_{\text{adv}}(\theta^t)||)^2}{2} m_p \mathcal{L}_{\text{adv}}(\theta^t) \\
&\leq \exp\left(-\eta\frac{||\nabla\mathcal{L}_{\text{adv}}(\theta^t)||_2^2}{\mathcal{L}_{\text{adv}}(\theta^t)} + m_p\frac{\eta^2}{2}||\nabla\mathcal{L}_{\text{adv}}(\theta^t)||_2^2\right) \\
&\leq \exp\left(-\eta\gamma_{2,q}||\nabla\mathcal{L}_{\text{adv}}(\theta^t)||_2 + m_p\frac{\eta^2}{2}||\nabla\mathcal{L}_{\text{adv}}(\theta^t)||_2^2\right).
\end{aligned}$$

where the last inequality holds by Lemma C.1.

Applying the previous inequality recursively from $s = 1$ to $t - 1$, we have

$$\mathcal{L}_{\text{adv}}(\theta^t) \leq \exp\left(-\eta\gamma_{2,q}\sum_{s=1}^{t-1}||\nabla\mathcal{L}_{\text{adv}}(\theta^s)||_2 + \sum_{s=1}^{t-1} m_p\frac{\eta^2}{2}||\nabla\mathcal{L}_{\text{adv}}(\theta^s)||_2^2\right).$$

Now since in the proof of Theorem 3.3 we showed that $\eta m_p < 1$ (35), combining the above inequality this with (37), we have

$$\sum_{s=1}^{t-1} m_p\frac{\eta^2}{2}||\nabla\mathcal{L}_{\text{adv}}(\theta^s)||_2^2 = \sum_{s=1}^{t-1} \frac{\eta}{2}||\nabla\mathcal{L}_{\text{adv}}(\theta^s)||_2^2 = \mathcal{L}_{\text{adv}}(\theta^1) - \mathcal{L}_{\text{adv}}(\theta^t) \leq \mathcal{L}_{\text{adv}}(\theta^1).$$

Combining this inequality with the upper bound on $\mathcal{L}_{\text{adv}}(\theta^1)$ in (34), we have

$$\mathcal{L}_{\text{adv}}(\theta^t) \leq \exp\left(-\eta\gamma_{2,q}\sum_{s=0}^{t-1}||\nabla\mathcal{L}_{\text{adv}}(\theta^s)||_2 - \gamma_{2,q}^2 + (1 + c\sqrt{d})\right).$$

Now for all $i \in [n]$, we have:

$$\exp\left(-\min_{||\delta_i||_q \leq c} y_i(x_i + \delta_i)^\top \theta^t\right) \leq n\exp\left(-\eta\gamma_{2,q}\sum_{s=0}^{t-1}||\nabla\mathcal{L}_{\text{adv}}(\theta^s)||_2 - \gamma_{2,q}^2 + (1 + c\sqrt{d})\right),$$

which yields
$$\min_{||\delta_i||_q \leq c} y_i(x_i + \delta_i)^\top \theta^t \geq \eta\gamma_{2,q} \sum_{s=0}^{t-1} ||\nabla\mathcal{L}_{\mathrm{adv}}(\theta^s)||_2 + \gamma_{2,q}^2 - (1 + c\sqrt{d}) - \log n.$$
Dividing both sides by $||\theta||_2$, and since $\lim_{t\to\infty} \mathcal{L}_{\mathrm{adv}}(\theta^t) = 0$, we have $\lim_{t\to\infty} ||\theta^t||_2 = \infty$. Hence,

$$\lim_{t\to\infty} \min_{||\delta_i||_q \leq c} y_i(x_i + \delta_i)^\top \frac{\theta^t}{||\theta^t||_2} \geq \lim_{t\to\infty} \eta\gamma_{2,q} \sum_{s=0}^{t-1} \frac{||\nabla\mathcal{L}_{\mathrm{adv}}(\theta^s)||_2}{||\theta^t||_2} - \frac{1 + c\sqrt{d} + \log n}{||\theta^t||_2} \qquad (46)$$
$$\geq \gamma_{2,q},$$
where the last inequality holds by $||\theta^t||_2 \leq \eta \sum_{s=0}^{t-1} ||\nabla\mathcal{L}_{\mathrm{adv}}(\theta^s)||_2$.

Hence in summary, we have
$$\min_{||\delta_i||_q \leq c} y_i(x_i + \delta_i)^\top \lim_{t\to\infty} \frac{\theta^t}{||\theta^t||_2} \geq \gamma_{2,q}.$$

Hence, we have $\lim_{t\to\infty} \theta^t/||\theta^t||_2$ is a solution to (10), but notice that the solution to (10) is unique since a multiple of its optimal solution would be the solution to (8) that
$$\min_{\theta\in\mathbb{R}^d} \frac{1}{2}||\theta||_2^2 \quad \text{s.t.} \quad \min_{\delta_i\in\Delta_i(q)} y_i(x_i + \delta_i)^\top \theta \geq 1, \forall i = 1, \dots, n,$$
which is a convex program with strongly convex objective. By this fact, we conclude that $\lim_{t\to\infty} \frac{\theta^t}{||\theta^t||_2} = u_{2,q}$. To further get the rate of convergence, we use the convergence of adversarial risk in (39), and establish the lower bound on $||\theta^t||_2$: $||\theta^t||_2 = \Omega(\log t)$. Combining this with (46), the claim follows immediately. $\qquad\square$

## D $\quad \ell_\infty$-NORM PERTURBATION

Recall that the robust SVM against $\ell_\infty$-norm perturbation parameterized by $c$ is formulated as
$$\gamma_{2,\infty} = \max_\theta \min_{i=1,\dots,n} \min_{||\delta_i||_\infty \leq c} \frac{y_i(x_i + \delta_i)^\top\theta}{||\theta||_2}, \qquad (47)$$
and its associated max-margin classifier is
$$u_{2,\infty} = \operatorname*{argmax}_{||\theta||_2=1} \min_{i=1,\dots,n} \min_{||\delta_i||_\infty \leq c} y_i(x_i + \delta_i)^\top\theta.$$
It is easy to see that for $c < \gamma_\infty$, both $\gamma_{2,\infty}$ and $u_{2,\infty}$ are well defined, and $\gamma_{2,\infty} > 0$.

Before showing parameter convergence, we first prove that the adversarial risk goes to zero. To avoid analyzing $\ell_\infty$-perturbation directly, which can go messy. For $\lambda > 0$, we define a smooth approximation of $\ell_1$-norm that
$$h_\lambda(\theta_j) = \sqrt{\theta_j^2 + \lambda}, \quad \text{and} \quad H_\lambda(\theta_j) = \sum_{j=1}^d h_\lambda(\theta_j).$$
Note that as $\lambda \to 0$, $H_\lambda(\theta) \to ||\theta||_1$ uniformly. We then define a smoothified version of (47) that we let perturbation set be $\Delta_i(\lambda) = \{\delta : \forall j \in [d], |\delta_j| \leq c\frac{h_\lambda(\theta_j)}{|\theta_j|}\}$, and the corresponding $\gamma_{2,\infty}$ and $u_{2,\infty}$ become
$$\gamma_{2,\lambda} = \max_\theta \min_{i=1,\dots,n} \min_{\delta_i\in\Delta_i(\lambda)} \frac{y_i(x_i + \delta_i)^\top\theta}{||\theta||_2}, \qquad (48)$$
$$u_{2,\lambda} = \operatorname*{argmax}_{||\theta||_2=1} \min_{i=1,\dots,n} \min_{\delta_i\in\Delta_i(\lambda)} y_i(x_i + \delta_i)^\top\theta. \qquad (49)$$
Note that the Hausdorff distance between $\Delta_i(\lambda)$ and $\{\delta : ||\delta||_\infty \leq c\}$ converges to 0 as $\lambda$ goes to 0. It can be seen that when $\lambda \to 0$, the smoothified problem (48) reduces to (47). That is, $\lim_{\lambda\to 0} \gamma_{2,\lambda} = \gamma_{2,\infty}$ and $\lim_{\lambda\to 0} u_{2,\lambda} = u_{2,\infty}$.

**Theorem D.1.** *Let perturbation set be* $\Delta_i(\lambda) = \{\delta : \forall j \in [d], |\delta_j| \leq c\frac{h_\lambda(\theta_j)}{|\theta_j|}\}$, *and let its associated adversarial risk be*
$$\mathcal{L}_{\mathrm{adv}}(\theta) = \frac{1}{n} \sum_{i=1}^n \max_{\delta_i\in\Delta_i(\lambda)} \exp\left(-y_i(x_i + \delta_i)^\top\theta\right).$$
*For* $c < \gamma_{2,\lambda}$, *letting* $\eta = \frac{1}{(1+2c\lambda^{-1/2})^2}$, *we have*
$$\mathcal{L}_{\mathrm{adv}}(\theta^t) \leq \mathcal{O}\left(\frac{\log^2 t(1 + 2c\lambda^{-1/2})^2}{t\gamma_{2,\lambda}}\right).$$

*Proof.* By the definition of perturbation set that $\Delta_i = \{\delta : \forall j \in [d], |\delta_j| \le c\frac{h_\lambda(\theta_j)}{|\theta_j|}\}$, we have

$$\mathcal{L}_{\mathrm{adv}}(\theta) = \frac{1}{n}\sum_{i=1}^n \exp\left(-y_i x_i^\top \theta + cH_\lambda(\theta)\right).$$

By some simple calculation, we have

$$\nabla H_\lambda(\theta) = \left(\frac{\theta_1}{\sqrt{\theta_1^2 + \lambda}}, \dots, \frac{\theta_d}{\sqrt{\theta_d^2 + \lambda}}\right), \nabla^2 H_\lambda(\theta) = diag\left(\frac{\lambda}{(\theta_1^2 + \lambda)^{3/2}}, \dots, \frac{\lambda}{(\theta_d^2 + \lambda)^{3/2}}\right).$$

Then, it holds that

$$\nabla \mathcal{L}_{\mathrm{adv}}(\theta) = \frac{1}{n}\sum_{i=1}^n \exp\left(-z_i^\top \theta + cH_\lambda(\theta)\right)\left(-z_i + c\nabla H_\lambda(\theta)\right),$$

$$\nabla^2 \mathcal{L}_{\mathrm{adv}}(\theta) = \frac{1}{n}\sum_{i=1}^n \exp\left(-z_i^\top \theta + cH_\lambda(\theta)\right)\left(z_i z_i^\top + c^2 \nabla H_\lambda(\theta)\nabla H_\lambda(\theta)^\top - 2z_i^\top \nabla H_\lambda(\theta) + c\nabla^2 H_\lambda(\theta)\right).$$

It can be verified that $\nabla^2 \mathcal{L}_{\mathrm{adv}}(\theta) \le (1 + \frac{2c}{\sqrt{\lambda}})^2 \mathcal{L}_{\mathrm{adv}}(\theta)I$. By Talyer expansion, we have

$$\mathcal{L}_{\mathrm{adv}}(\theta^{t+1}) \le \mathcal{L}_{\mathrm{adv}}(\theta^t) - \eta\|\nabla \mathcal{L}_{\mathrm{adv}}(\theta^t)\|_2^2 + (1 + \frac{2c}{\sqrt{\lambda}})^2\frac{\eta^2}{2}\max\{\mathcal{L}_{\mathrm{adv}}(\theta^t), \mathcal{L}_{\mathrm{adv}}(\theta^{t+1})\}\|\nabla \mathcal{L}_{\mathrm{adv}}(\theta^t)\|_2^2. \tag{50}$$

Now we show that $\mathcal{L}_{\mathrm{adv}}(\theta^{t+1}) \ge \mathcal{L}_{\mathrm{adv}}(\theta^t)$ does not hold when $\eta \le \frac{1}{(1+2c\lambda^{-1/2})^2 \mathcal{L}_{\mathrm{adv}}(\theta^t)}$. Suppose the contrary holds. By (50), we have

$$\mathcal{L}_{\mathrm{adv}}(\theta^{t+1}) \le \left(1 - \frac{\eta^2\|\nabla \mathcal{L}_{\mathrm{adv}}(\theta^t)\|_2^2}{2}(1 + \frac{2c}{\sqrt{\lambda}})^2\right)^{-1}\left(\mathcal{L}_{\mathrm{adv}}(\theta^t) - \eta\|\nabla \mathcal{L}_{\mathrm{adv}}(\theta^t)\|_2^2\right) < \mathcal{L}_{\mathrm{adv}}(\theta^t).$$

where the last inequality holds by $\eta = \frac{1}{(1+2c\lambda^{-1/2})^2 \mathcal{L}_{\mathrm{adv}}(\theta^t)}$. Hence we obtain a contradiction.

Note that $\mathcal{L}_{\mathrm{adv}}(\theta^0) = 1$, and if $\eta \le \frac{1}{(1+2c\lambda^{-1/2})^2}$, $\eta \le \frac{1}{(1+2c\lambda^{-1/2})^2 \mathcal{L}_{\mathrm{adv}}(\theta^t)}$ holds for $t = 0$, and $\mathcal{L}_{\mathrm{adv}}(\theta^1) \le 1$. Consequently, we can inductively show that $\mathcal{L}_{\mathrm{adv}}(\theta^t) \le 1$ for all $t$, and $\eta \le \frac{1}{(1+2c\lambda^{-1/2})^2 \mathcal{L}_{\mathrm{adv}}(\theta^t)}$ always holds if we let $\eta = \frac{1}{(1+2c\lambda^{-1/2})^2}$.

By the choice of $\eta$, we obtain the following recursion taht

$$\mathcal{L}_{\mathrm{adv}}(\theta^{t+1}) \le \mathcal{L}_{\mathrm{adv}}(\theta^t) - \eta\|\nabla \mathcal{L}_{\mathrm{adv}}(\theta^t)\|_2^2 + (1 + \frac{2c}{\sqrt{\lambda}})^2\frac{\eta^2 \mathcal{L}_{\mathrm{adv}}(\theta^t)}{2}\|\nabla \mathcal{L}_{\mathrm{adv}}(\theta^t)\|_2^2 \tag{51}$$

$$= \mathcal{L}_{\mathrm{adv}}(\theta^t) - \frac{\eta}{2}\|\nabla \mathcal{L}_{\mathrm{adv}}(\theta^t)\|_2^2. \tag{52}$$

Using the previous recursion we have that for any $\theta \in \mathbb{R}^d$,

$$\|\theta^{t+1} - \theta\|_2^2 = \|\theta^t - \theta\|_2^2 - 2\eta\left\langle \nabla \mathcal{L}_{\mathrm{adv}}(\theta^t), \theta^t - \theta\right\rangle + \eta^2\|\nabla \mathcal{L}_{\mathrm{adv}}(\theta^t)\|_2^2$$

$$\le \|\theta^t - \theta\|_2^2 - 2\eta\left(\mathcal{L}_{\mathrm{adv}}(\theta^t) - \mathcal{L}_{\mathrm{adv}}(\theta)\right) + 2\eta\left(\mathcal{L}_{\mathrm{adv}}(\theta^t) - \mathcal{L}_{\mathrm{adv}}(\theta^{t+1})\right)$$

$$= \|\theta^t - \theta\|_2^2 - 2\eta\left(\mathcal{L}_{\mathrm{adv}}(\theta^{t+1}) - \mathcal{L}_{\mathrm{adv}}(\theta)\right),$$

where the second inequality holds by convexity and (51). Summing up the previous inequality from $s = 0$ to $s = t - 1$ and by $\mathcal{L}_{\mathrm{adv}}(\theta^{s+1}) \le \mathcal{L}_{\mathrm{adv}}(\theta^s)$, we have

$$\mathcal{L}_{\mathrm{adv}}(\theta^t) - \mathcal{L}_{\mathrm{adv}}(\theta) \le \frac{1}{2t\eta}\|\theta\|_2^2.$$

Taking $\theta = \frac{\log t}{\gamma_{2,\lambda}}u_{2,\lambda}$, we have

$$\mathcal{L}_{\mathrm{adv}}(\theta) = \frac{1}{n}\sum_{i=1}^n \max_{\delta_i \in \Delta_i(\lambda)} \exp\left(-y_i(x_i + \delta_i)^\top \theta\right)$$

$$= \frac{1}{n}\sum_{i=1}^n \max_{\delta_i \in \Delta_i(\lambda)} \exp\left(-y_i(x_i + \delta_i)^\top \frac{\log t}{\gamma_{2,\lambda}}u_{2,\lambda}\right) \le \frac{1}{t}.$$

where the last inequality holds by $\max_{\delta_i \in \Delta_i} y_i(x_i + \delta_i)^\top u_{2,\lambda} \ge \gamma_{2,\lambda}$. Hence we obtain

$$\mathcal{L}_{\mathrm{adv}}(\theta^t) \le \frac{1}{t} + \frac{\log^2 t}{t\gamma_{2,\lambda}\eta} = \mathcal{O}\left(\frac{\log^2 t(1 + 2c\lambda^{-1/2})^2}{t\gamma_{2,\lambda}}\right).$$

$\square$

Before showing parameter convergence, we need the following lemma which is a generalization of Lemma 10 in Gunasekar et al. (2018a), but with much simpler proof.

**Lemma D.1.** *Fix $c < \gamma_{2,\lambda}$, for any $\theta \in RR^d$, we have*
$$||\nabla\mathcal{L}_{adv}(\theta)||_2 \geq \mathcal{L}_{adv}(\theta)\gamma_{2,\lambda}.$$

*Proof.*
$$-\nabla\mathcal{L}_{adv}(\theta) = \frac{1}{n}\sum_{i=1}^{n}\exp\left(-y_i\widetilde{x}_i\right)y_i\widetilde{x}_i.$$

where $\widetilde{x}_i = \operatorname{argmin}_{x'_i - x_i \in \Delta_i(\lambda)} y_i(x'_i)^\top\theta$. Then by the definition of $\gamma_{2,\lambda}$ and $u_{2,\lambda}$ (48), we have
$$\langle y_i\widetilde{x}_i, u_{2,\lambda}\rangle \geq \gamma_{2,\lambda}$$

From which we obtain $\langle -\nabla\mathcal{L}_{adv}(\theta), u_{2,\lambda}\rangle \geq \mathcal{L}_{adv}(\theta)\gamma_{2,\lambda}$, the claim follows by Cauchy-Schwarz inequality. □

**Theorem D.2.** *Under the same setting as in Theorem D.1, we have*
$$\lim_{t\to\infty}\frac{\theta^t}{||\theta^t||_2} = u_{2,\lambda}.$$

*Proof.* Recall that in Theorem D.1 we showed in (51) that

$$\mathcal{L}_{adv}(\theta^{t+1}) \leq \mathcal{L}_{adv}(\theta^t) - \eta||\nabla\mathcal{L}_{adv}(\theta^t)||_2^2 + (1 + \frac{2c}{\sqrt{\lambda}})^2\frac{\eta^2\mathcal{L}_{adv}(\theta^t)}{2}||\nabla\mathcal{L}_{adv}(\theta^t)||_2^2$$

$$\leq \exp\left(-\eta\frac{||\nabla\mathcal{L}_{adv}(\theta^t)||_2^2}{\mathcal{L}_{adv}(\theta^t)} + (1 + \frac{2c}{\sqrt{\lambda}})^2\frac{\eta^2}{2}||\nabla\mathcal{L}_{adv}(\theta^t)||_2^2\right)$$

$$\leq \exp\left(-\eta\gamma_{2,\lambda}||\nabla\mathcal{L}_{adv}(\theta^t)||_2 + (1 + \frac{2c}{\sqrt{\lambda}})^2\frac{\eta^2}{2}||\nabla\mathcal{L}_{adv}(\theta^t)||_2^2\right),$$

where the last inequality holds by Lemma D.1. Applying the previous inequality recursively from $s = 0$ to $t - 1$, we have

$$\mathcal{L}_{adv}(\theta^t) \leq \exp\left(-\eta\gamma_{2,\lambda}\sum_{t=0}^{t-1}||\nabla\mathcal{L}_{adv}(\theta^s)||_2 + \sum_{s=0}^{t-1}(1 + \frac{2c}{\sqrt{\lambda}})^2\frac{\eta^2}{2}||\nabla\mathcal{L}_{adv}(\theta^s)||_2^2\right).$$

Now by (51), we have

$$\sum_{s=0}^{t-1}(1 + \frac{2c}{\sqrt{\lambda}})^2\frac{\eta^2}{2}||\nabla\mathcal{L}_{adv}(\theta^s)||_2^2 = \sum_{s=0}^{t-1}\frac{\eta}{2}||\nabla\mathcal{L}_{adv}(\theta^s)||_2^2 = \mathcal{L}_{adv}(\theta^0) - \mathcal{L}_{adv}(\theta^t) \leq 1,$$

which yields

$$\mathcal{L}_{adv}(\theta^t) \leq \exp\left(-\eta\gamma_{2,\lambda}\sum_{s=0}^{t-1}||\nabla\mathcal{L}_{adv}(\theta^s)||_2 + 1\right).$$

Next for all $i \in [n]$, we have

$$\exp\left(-\min_{\delta_i\in\Delta_i(\lambda)}y_i(x_i + \delta_i)^\top\theta^t\right) = \exp\left(-y_ix_i^\top\theta + cH_\lambda(\theta^t)\right)$$

$$\leq n\exp\left(-\eta\gamma_{2,\lambda}\sum_{s=0}^{t-1}||\nabla\mathcal{L}_{adv}(\theta^s)||_2 + 1\right),$$

which implies

$$\min_{\delta_i\in\Delta_i(\lambda)}y_i(x_i + \delta_i)^\top\theta^t \geq \eta\gamma_{2,\lambda}\sum_{s=0}^{t-1}||\nabla\mathcal{L}_{adv}(\theta^s)||_2 - 1 - \log n.$$

Dividing both sides by $||\theta||_2$, and since $\lim_{t\to\infty}\mathcal{L}_{adv}(\theta^t) = 0$, we have $\lim_{t\to\infty}||\theta^t||_2 = \infty$. Hence,

$$\lim_{t\to\infty}\min_{\delta_i\in\Delta_i(\lambda)}y_i(x_i + \delta_i)^\top\frac{\theta^t}{||\theta^t||_2} \geq \lim_{t\to\infty}\eta\gamma_{2,\lambda}\sum_{s=0}^{t-1}\frac{||\nabla\mathcal{L}_{adv}(\theta^s)||_2}{||\theta^t||_2} - \frac{1 + \log n}{||\theta^t||_2} \geq \gamma_{2,\lambda},$$

where the last inequality holds by $||\theta^t||_2 \leq \eta\sum_{s=0}^{t-1}||\nabla\mathcal{L}_{adv}(\theta^s)||_2$.

In summary, we have

$$\min_{\delta_i \in \Delta_i(\lambda)} y_i(x_i + \delta_i)^\top \lim_{t \to \infty} \frac{\theta^t}{||\theta^t||_2} \geq \gamma_{2,\lambda}.$$

Hence $\lim_{\theta^t} ||\theta^t||_2$ is a solution to (48). Note that the solution to (48) is unique since it is equivalent to

$$\min_{\theta \in \mathbb{R}^d} \frac{1}{2} ||\theta||_2^2 \quad \text{s.t.} \quad \min_{\delta_i \in \Delta_i(\lambda)} y_i(x_i + \delta_i)^\top \theta \geq 1, \forall i = 1, \ldots, n.$$

We thus conclude that $\lim_{t \to \infty} \frac{\theta^t}{||\theta^t||_2} = u_{2,\lambda}$. $\qquad \square$

To summarize, we have shown that for all $\lambda > 0$, $\lim_{t \to \infty} \frac{\theta^t}{||\theta^t||_2} = u_{2,\lambda}$. The $\ell_\infty$-norm perturbation corresponds to the case when $\lambda \to 0$, it is natural to conclude that for $\ell_\infty$ perturbation, we have $\lim_{t \to \infty} \frac{\theta^t}{||\theta^t||_2} = u_{2,\infty}$. The discussion for $q = 1$ follows similar argument, hence we omit the details here.

## E    ADDITIONAL EXPERIMENTS ON PERTURBATION LEVEL AND SPEED-UP

We provide additional experiments on the connection of perturbation level $c$ and the speed-up effect of adversarial training for neural networkds. We run GDAT with $\ell_\infty$-norm perturbation. The setup of the experiments is exactly the same as the setup in Section 4. We will vary the perturbation level $c$ used in GDAT algorithm in $\{0.1, 0.15, 0.2\}$.

From Figure 3 we could see that GDAT indeed accelerates convergence of loss and accuracy on clean training samples. Moreoever, the acceleration effect is stronger when we use larger perturbation level, and this relationship is consistent across different width of hidden layer.

Similar speed-up effects on the test loss and test accuracy evaluated on clean test samples are also observed for GDAT. From Figure 4, we see that the speed-up effects become stronger when we use larger perturbation level, and this relationship is consistent across different width of hidden layer. Traditionally, the benefit of adversarial training is understood as two fold: 1. it improves the robustness of the learning algorithm, i.e., the solution has better loss toward adversarilly perturbed seample; 2. it has better generalization ability. Our experiments demonstrate a third property of adverserial training that is not known in literature before, i.e., adversarial training accelerates convergence.

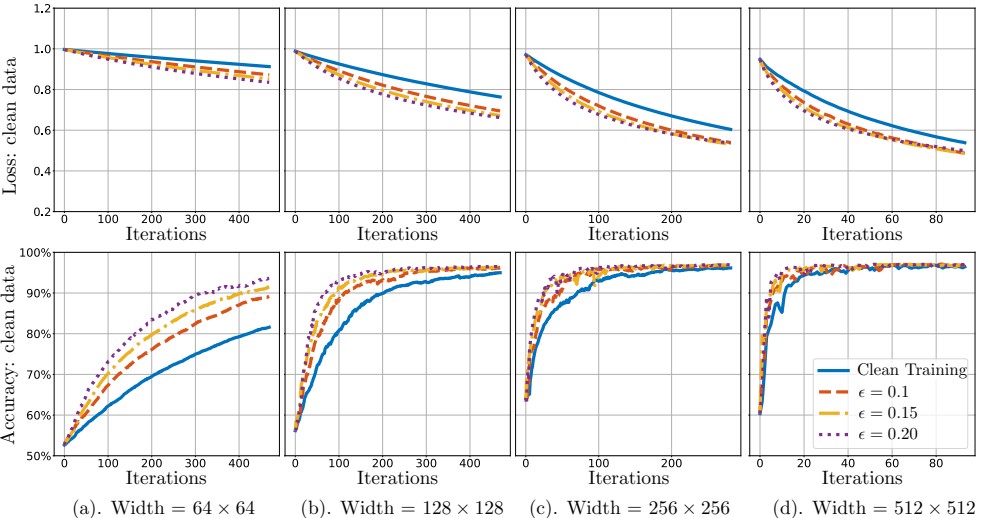

Figure 3: GDAT with Different Perturbation Level: Clean Training Loss

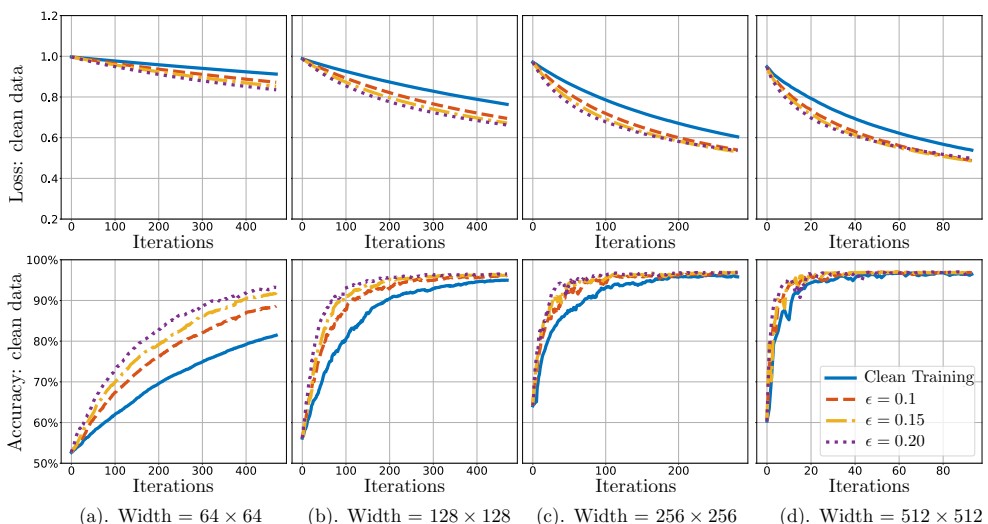

Figure 4: GDAT with Different Perturbation Level: Clean Test Loss

