# OpenReview forum: "Implicit Bias of Gradient Descent based Adversarial Training on Separable Data"
_ICLR.cc/2020/Conference — Accept (Poster)_

### Official Review · AnonReviewer1 · 2019-10-20
**Official Blind Review #1**

**Rating:** 3

**Review:**

The paper applies theory on implicit bias of gradient descent to an adversarial training toy example. It attempts to utilize the theoretical results for deriving insights on how adversarial training establishes robustness. The theoretical results are complemented with experiments on linear classifiers and small neural networks.

The key technical contributions look mostly like an application of the existing theory on implicit bias of gradient descent, thus I would rate the originality of this work as moderate. From the point of view of adversarial machine learning, I don't consider the results to be significant. In particular, there is a large discrepancy between the theoretical results on the toy example and the empirical convergence behaviour of adversarial training of neural networks (compare Figure 2 and 3). In particular, the theoretically derived exponential speed-up of adversarial training for the toy example is not reflected by empirical observations for practically relevant adversarial training tasks (e.g. consider the convergence behaviours reported by Goodfellow et al. (2014) or Madry et al. (2017)). Thus, the theory doesn't seem to be generally applicable beyond the linear toy model. A clear discussion of its limitations is missing.

Overall, while this may be an interesting extension of the existing body of literature on the implicit bias of gradient descent, I find that from an adversarial machine learning perspective this paper does not meet its objective to derive generally applicable insights on how adversarial training works.

**Experience Assessment:**

I have published one or two papers in this area.

**Review Assessment: Checking Correctness Of Derivations And Theory:**

I did not assess the derivations or theory.

**Review Assessment: Checking Correctness Of Experiments:**

I assessed the sensibility of the experiments.

**Review Assessment: Thoroughness In Paper Reading:**

I made a quick assessment of this paper.

---

> ### Author Response · Authors · 2019-11-15
> **To Reviewer #1**
>
> Thank you for the valuable feedback.
>
> We first would like to emphasize that our technical results, though inspired by existing theories on the implicit bias of gradient descent, take significant efforts and require us to develop new techniques tailored for the adversarial learning setting. Specifically, Lemma $3.1$, Lemma C$.1$, and their applications in our proofs are generalizations of a key step in the convergence analysis of the Perceptron algorithm in the adversarial training setting, which are unknown in the existing literature to the best of our knowledge. Our lemma significantly improves upon [1], of which the proof applies sophisticated tools from convex duality, and only applies to clean training. In contrast, our lemmas admit a much simpler, intuitive proof, and can be applied to much broader settings. Our acceleration result for GDAT with $\ell_2$ norm perturbation is, to the best of our knowledge, the first result that shows the acceleration effect of adversarial training. The analysis not only requires a characterization that shows the boundedness of the iterate on orthogonal space of the $\ell_2$ norm max-margin classifier in the adversarial training setting, but also demonstrates accelerated progress on the direction of the $\ell_2$ norm max-margin classifier. Both of these elements are not available from existing results. In addition,  we prove that GDAT with different types of perturbation converge to different maximum mixed-norm margin classifiers. This is also the first result showing that the gradient descent based adversarial training indeed adapts to adversary geometry and provide robustness.
>
> For practically used neural networks, there are many factors coupled together: batch normalization, dropout, skip-layer connections, etc. It is important to note there is still a lack of understandings on many of the aforementioned techniques, let alone with their possible interactions with adversarial training. With all the confounding factors, the effect of using GDAT as the training algorithm would be hard to identify.  We conduct experiments on simple neural network models to focus on the effect of GDAT, and empirically observe its acceleration effect. Please also kindly refer to Appendix E where we conduct more experiments with various perturbation levels. We observe that as the perturbation level increases, the acceleration effect becomes more obvious for networks of all sizes. This phenomenon shows that the acceleration effect of GDAT might depend on the network size, which we believe is worthy of future study.
>
> Our experiments on neural networks focus on the convergence behavior of the clean training accuracy using GDAT as the training algorithm. Our focus largely differs from most existing literature, which puts more emphasis on comparing the final clean accuracy of adversarial training with clean training. To the best of our knowledge, [2] and [3] all consider the final clean accuracy of the trained network, instead of the convergence behavior of the clean accuracy. Interestingly, the adversarial training approach has been applied to natural language processing task, and shows improvement in terms of clean accuracy [5]. In addition, adversarial training in reinforcement learning has also been observed to speed up the learning process [6]. Based on these observations, we believe that the potential of adversarial training is not only limited to promoting the robustness of the learned model, and we are in need of further studies on its computational and generalization properties.
>
> Finally, we would like to emphasize that our results serve as the first step towards understanding the computational properties of the adversarial training, and certainly not a solution once and for all. Our future work would focus on understanding the computational properties of adversarial training for nonlinear networks.
>
>
> [1][Gunasekar, S., et al.  ``Characterizing Implicit Bias in Terms of Optimization Geometry](https://arxiv.org/abs/1802.08246)
> [2][Goodfellow, I., Shlens J., and Szegedy, C. ``Explaining and Harnessing Adversarial Examples](https://arxiv.org/abs/1412.6572)
> [3][Madry, A., et al. ``Towards Deep Learning Models Resistant to Adversarial Attacks](https://arxiv.org/pdf/1706.06083)
> [4][Shafahi, A., et al. `` Adversarial Training for Free!](https://arxiv.org/abs/1904.12843)
> [5][Zhu, C., et al.  ``FreeLB: Enhanced Adversarial Training for Language Understanding](https://arxiv.org/abs/1909.11764)
> [6][Pinto, L., et al. ``Robust Adversarial Reinforcement Learning](https://arxiv.org/abs/1703.02702)

---

> > ### Comment · AnonReviewer1 · 2019-11-15
> > **From an adversarial machine learning perspective, I'm still not too enthusiastic.**
> >
> > Thank you for the response.
> >
> > I have no doubts that deriving the results required - as the authors indicate in their response - significant efforts and the development of novel techniques. My main reservation is that, while this work advances the state-of-the-art in the theory of implicit bias of gradient descent, I don't find it contributes to a better understanding of the workings of adversarial training, which is what I'm mostly interested in. The theoretical results on a simple toy model don't seem to be reconcilable with practical observations on the convergence of adversarial training, including the experiment conducted by the authors.
> >
> > For someone deeply interested in the theory of implicit bias of gradient descent, this may be a very interesting and insightful contribution.

---

### Official Review · AnonReviewer3 · 2019-10-22
**Official Blind Review #3**

**Rating:** 8

**Review:**

The paper studies the implicit bias of gradient based adversarial training for linear models on separable data with the exponential loss. Both l2 and general lq perturbations are studied.

For l2, the authors show that for small enough perturbations, the algorithm converges in direction to the max l2 margin solution, with faster convergence rates compared to standard gradient descent, both for the clean loss and the parameter direction (O(1/sqrt(T)) instead of O(1/log(T))).
For lq, convergence to a different max-margin direction is shown, given according to a mixture of the l2 and lp norms, though the parameter convergence is slower.
These results are further illustrated by numerical experiments.

The topic of the paper is novel and interesting, in particular the finding that adversarial training can lead to benefits in terms of optimization in addition to robustness, as well as the characterization of the inductive bias obtained when using lq perturbations in adversarial training (i.e. a mixture of lp and l2 norms, rather than just lp). Granted, the setting of linear models is a bit limited, but it provides a first step for more realistic models. The paper is also well written, and also has a nice numerical validation of the results. Overall, I am in favor of acceptance.

Here are some questions/comments that could be further discussed:
* for l2, while the obtained rates for parameter convergence are better in their dependence on T, they depend on a data-dependent quantity alpha in contrast to standard GD, suggesting that the rate could be worse for small alpha: is there a trade-off here? would standard GD be preferable if alpha is small, or could the current rate (7) automatically adapt to such settings?

* in Theorem 3.4, how does the choice of c come into play? can you obtain better rates by optimizing it (ideally this should happen when q=2)? Regarding the comment on tending to the lq-norm margin for c -> gamma_q, is this at the cost of poorer convergence?

* for lq perturbations, while I agree that the studied algorithm is interesting since that's what's used in practice for deep networks, I do wonder if in this specific setting you could get better convergence (and with the right norm lp instead of the mixture) by optimizing using the appropriate geometry, e.g. with mirror descent.

* the analyzed algorithm considers optimal perturbations -- do you have a sense of how robust the theory is to errors in the inner optimization?


minor comments/typos:
* second bullet in 'main contributions': 'mixed-norm margin' could be further explained, or at least point to the definition. The terminology is also confusing as it sounds like a matrix mixed-norm
* 'polyhedral cone ...': bar{u} should be u?
* after Theorem 3.2 'not an issue': not so clear, this should be discussed further (see comment above)
* section 3.2, "defer discussion for 1, infty": you could provide a flavor of the results for these cases in the main text, given their prominence in practice
* figure 1(b): for the direction plot, is it actually u_2? how should this plot be interpreted given that the two curves converge to a different direction?

**Experience Assessment:**

I have read many papers in this area.

**Review Assessment: Checking Correctness Of Derivations And Theory:**

I assessed the sensibility of the derivations and theory.

**Review Assessment: Checking Correctness Of Experiments:**

I assessed the sensibility of the experiments.

**Review Assessment: Thoroughness In Paper Reading:**

I read the paper at least twice and used my best judgement in assessing the paper.

---

> ### Author Response · Authors · 2019-11-15
> **To Reviewer #3**
>
> Thank you for your valuable feedback, we will update the typos accordingly and further polish our presentations.
>
> Major Comments:
>
> (1). We emphasize here that data dependence is one of the most fundamental traits of the optimization algorithm. In fact, the rate of the gradient descent algorithm converging to the max-margin classifier is also data-dependent [1]. Specifically, it depends on the minimal margin value gap between the support vectors and the non-support vectors. Depending on the analysis, the data dependence could be reflected in the final convergence rate in different forms. Here we measure the data dependence in terms of the defined $\alpha$ in Lemma 3.2, which is different from [1].
>
> (2). For Theorem 3.4, changing $c$ first changes the set of stepsize for which GDAT guarantees to converge. For $q=2$, we can provide an optimal choice of $c$ due to the fact that $u_{2,2}(c) = u_2$ for all  $c < \gamma_2$. For other values of $q$, we note that the converged classifier $u_{2,q}(c)$ in fact depends on $c$, so setting different values of $c$ will result in different classifiers. Hence the problem of choosing the optimal $c$ is not a particularly well-defined problem.
>
> (3). Investigating the convergence rate of mirror descent adapted to different adversary geometry is a great question from our perspective. We believe that more sophisticated techniques have to be developed for this case, and we leave this as one of our future directions.
>
> (4). For linear models, inner optimization problems can be solved exactly. For the approximately solved inner maximization problem, we believe that the acceleration effect for $\ell_2$ norm perturbation still holds. In addition, for GDAT with general $\ell_q$ norm (including $\ell_2$ norm) perturbations, we believe that the final converged classifier should be an approximation to the maximum mixed-norm margin classifier if the inner maximization problem is solved to sufficient precision. We plan to study these problems in detail in our followup works.
>
> (5) Minor Comments:
>
> Regarding Section 3.2 and Figure 1.(b): Our theoretical result holds for arbitrarily $q < \infty$. Note that as $q$ tends to infinity, the maximum mixed-norm margin classifier $u_{2,q}$ converges to $u_{2,\infty}$. To empirically validate this, we compare the convergence for GDAT with $\ell_\infty$ norm perturbation and GDAT with $\ell_q$ perturbation when $q$ is a large value ($1000$). The y-axis in Figure 1.(b) lower-left part should be $1- \langle \theta^t,u_{2,\infty}\rangle$, which we have revised in our updated draft.  Figure 1.(b) then demonstrates that the convergence behavior for $q = \infty$ and $q = 1000$ is extremely close.
>
> [1][Soudry, D., et al. ``The Implicit Bias of Gradient Descent on Separable Data](https://arxiv.org/abs/1710.10345)

---

### Official Review · AnonReviewer4 · 2019-10-23
**Official Blind Review #4**

**Rating:** 6

**Review:**

**Contributions:**
This paper extends results on the implicit bias of gradient descent (GD) Soudry et al. (2018)[3] for the case of gradient descent based adversarial training (GDAT).

**1) Theoretical result for L2 norm:** Convergence of standard risk in GDAT is significantly faster than its counterpart in the standard clean training using GD: For any fixed iteration T, when the adversarial perturbation during training has bounded l2-norm, the classifier learned by GDAT converges to the maximum l2-norm margin classifier at the rate of O(1/√T), exponentially faster than the rate O (1/ log T ) obtained when training with only cleaned data Soudry et al. (2018) [3].

**2) theoretical result for Lq norm:** When the adversarial perturbation during training has bounded l_q-norm with q > 1, with a proper choice of c, the gradient descent based adversarial training is directionally convergent to a maximum mixed-norm margin classifier, which has a natural interpretation of robustness, as being the maximum $l_2$-norm margin classifier under worst-case $l_q$-norm perturbation to the data.

The paper is well written and easy to understand. I haven’t checked the proofs, but I focused on readability and motivations.

I have the following questions.

**Theoretical questions:**
- Can you please explain the intuition of lemma 3.2 and 3.3 for convergence?
**Experiments questions:**
Keeping in mind that the core of the paper is about proving that GDAT enjoys the same implicit bias as GD and better convergence performance on clean data, I can’t help myself by asking you how these properties reflect in experiments with deep Neural Networks:
    - Can you please clarify your motivation for the reduction of the performance gap when the with of the hidden layer increases (Figure 2)? It is not clear what do you mean by ...“as network width increases, the margin on the samples outputted by the hidden layer also increases”...
My personal interpretation of the results is that the inner maximization problem is much harder to solve and being approximated, the effect of the GDAT is much less prominent, meaning that PGD is not enough to benefit from adversarial training in this sense. What do you think about it?
    - Take away message of the paper: adversarial training accelerates convergence. Theoretical results are for the training empirical loss.
Is this also the case of the validation loss? You observe this behavior for the case of a single MLP on MNIST binary problem 2 vs 9 (appendix E) claiming that adversarial training improves generalization performance.
This is not observed in the literature, it is actually the opposite [1, 2]: it exists a tension between robustness and performance on the clean test set. Why do you think you are not observing this? My guesses:
    - you are analysing a very simple binary problem
    - your model is very simple and comparable to a linear classifier, so your theoretical results hold.

[1][1805.12152 Robustness May Be at Odds with Accuracy](https://arxiv.org/abs/1805.12152)
[2][1810.12715 On the Effectiveness of Interval Bound Propagation for Training Verifiably Robust Models](https://arxiv.org/abs/1810.12715)
[3][1710.10345 The Implicit Bias of Gradient Descent on Separable Data](https://arxiv.org/abs/1710.10345)


**Experience Assessment:**

I have read many papers in this area.

**Review Assessment: Checking Correctness Of Derivations And Theory:**

I did not assess the derivations or theory.

**Review Assessment: Checking Correctness Of Experiments:**

I assessed the sensibility of the experiments.

**Review Assessment: Thoroughness In Paper Reading:**

I read the paper at least twice and used my best judgement in assessing the paper.

---

> ### Author Response · Authors · 2019-11-15
> **To Reviewer #4**
>
> Thank you for the valuable feedback. Below are our responses to your comments:
>
> Theoretical questions:
>
> (1). The intuition for Lemma 3.2 is that if $\alpha > 0$, then for any iteration there exists a support vector (we call it the primal vector here), whose update direction has positive inner product with  $\theta^t$'s projection onto the orthogonal space of the max-margin classifier. This implies that the update direction contributed from such primal vector effectively reduces the magnitude of $\theta^t$'s projection onto the orthogonal space of the max-margin classifier. Moreover, for all the other examples whose update directions tend to increase the magnitude of  $\theta^t$'s projection onto the orthogonal space of the max-margin classifier, their update strength is much smaller than the update strength of the primal vector. Hence the overall update direction is dominated by the primal vector. Then the overall magnitude of $\theta^t$'s projection onto the orthogonal space of the max-margin classifier remains bounded.
>
>
> (2). The intuition for Lemma 3.3 comes from our observations that even though GDAT minimizes the adversarial training loss with a slower convergence rate in comparison with the  clean training, when we try to measure the progress made on the direction of the max-margin classifier, this slower rate appears in a logarithmic term (see the denominator in Lemma 3.3). In contrast, since the magnitude of projection of $\theta^t$ onto the orthogonal space of the max-margin classifier remains bounded, we can associate the decrease in training loss directly with the growth of $\theta^t$ along the direction of the max-margin classifier, which gives us a factor of $1/(\gamma_2 -c )$ increase compared to the clean training (see the denominator in Lemma 3.3). Combine this with the previous observation gives us the proof ingredients of Lemma 3.3.
>
> Experiment questions:
>
> (1). For the reduction in the performance gap when the width of the hidden layer increases, our intuition for explanation is based on Theorem 3.2, which demonstrates that as the margin value increases, to retain similar acceleration effects, the desired perturbation strength $c$ should also increase. We believe that approximately solving the inner maximization problem can also affect the convergence behavior, but it might be not sufficient enough to explain the reduction of acceleration effect when the network size increases. We hypothesize that as the network size increases, the margin value (geometric margin) also increases, hence suggesting a larger perturbation level. Please kindly refer to Appendix~E for experiments with various perturbation levels. We observe that with a larger perturbation level, the speed-up effect of GDAT indeed increases. Interestingly, in the adversarial learning literature, there have also been empirical results suggesting this hypothesis. Specifically, [1] has reported that larger networks are more robust to adversarial attacks, implying a larger geometric margin for clean samples. We believe our neural network experiments demonstrate some interesting connections between the network size and adversarial perturbation level, which is worth of future investigations.
>
> (2). The acceleration effect also holds for the validation loss, as long as the training set and the test set share similar max-margin solutions. For neural network experiments, we observe that the convergence for clean training accuracy/loss is accelerated by GDAT for MNIST dataset. We do not claim that the final accuracy is improved by GDAT. In fact, our empirical results do not contradict with the previous theoretical investigations [1][2], as they focus on the comparison of final clean accuracies obtained by GDAT and clean training. We focus on the convergence of the clean accuracy during the training process. We believe the reason why we do not observe significant tension between robustness and accuracy here is that MNIST dataset are very close to being linearly separable. For general dataset, we expect that adversarial training could speed up the convergence of clean accuracies in the initial stage, and plateaus quicker with a lower final clean accuracy.
>
> [1][Madry, A., et al. ``Towards Deep Learning Models Resistant to Adversarial Attacks](https://arxiv.org/pdf/1706.06083)
> [2][Tsipras, D., et al.  ``Robustness May Be at Odds with Accuracy](https://arxiv.org/abs/1805.12152)
> [3][Gowal, S., et al.  ``On the Effectiveness of Interval Bound Propagation for Training Verifiably Robust Models](https://arxiv.org/abs/1810.12715)

---

### Decision · Program_Chairs · 2019-12-19

**Decision:**

Accept (Poster)

**Comment:**

This paper provides theoretical guarantees for adversarial training.  While the reviews raise a variety of criticisms (e.g., the results are under a variety of assumptions), overall the paper constitutes valuable progress on an emerging problem.